# Models of human preference for learning reward functions

**W. Bradley Knox**[*][†]   *bradknox@cs.utexas.edu*
*Bosch*
*The University of Texas at Austin*
*Google Research*

**Stephane Hatgis-Kessell**[*]   *stephane@cs.utexas.edu*
*The University of Texas at Austin*

**Serena Booth**   *sbooth@mit.edu*
*Bosch*
*MIT CSAIL*

**Scott Niekum**   *sniekum@cs.utexas.edu*
*The University of Texas at Austin*
*University of Massachusetts Amherst*

**Peter Stone**   *pstone@cs.utexas.edu*
*The University of Texas at Austin*
*Sony AI*

**Alessandro Allievi**   *alessandro.allievi@us.bosch.com*
*Bosch*

**Reviewed on OpenReview:** *https://openreview.net/forum?id=hpKJkVoThY*

Figure 1: Illustrations of segment pairs for which the common partial return preference model poorly explains intuitive human preference. The task has $-1$ reward each time step, penalizing time taken to reach the goal. In both pairs, both segments have the same partial return $(-2)$, but the one on the right is nonetheless the intuitive choice. Additionally, the right segment in each pair consists only of optimal actions, whereas the left segment includes at least one suboptimal action. Regret, which our proposed preference model is based upon, is designed to measure a segment's deviation from optimal decision making. The right segment in each pair is therefore more likely to be preferred by a regret preference model. In the left pair, the preferred segment has a higher end state value. In the right pair, the preferred segment has a lower start state value, indicating a lower opportunity cost (i.e., it did not waste a more valuable start state).

---

[*]The first two authors contributed equally.
[†]Work partially done while at Bosch and Google Research

## Abstract

The utility of reinforcement learning is limited by the alignment of reward functions with the interests of human stakeholders. One promising method for alignment is to learn the reward function from human-generated preferences between pairs of trajectory segments, a type of reinforcement learning from human feedback (RLHF). These human preferences are typically assumed to be informed solely by partial return, the sum of rewards along each segment. We find this assumption to be flawed and propose modeling human preferences instead as informed by each segment's regret, a measure of a segment's deviation from optimal decision-making. Given infinitely many preferences generated according to regret, we prove that we can identify a reward function equivalent to the reward function that generated those preferences, and we prove that the previous partial return model lacks this identifiability property in multiple contexts. We empirically show that our proposed regret preference model outperforms the partial return preference model with finite training data in otherwise the same setting. Additionally, we find that our proposed regret preference model better predicts real *human* preferences and also learns reward functions from these preferences that lead to policies that are better human-aligned. Overall, this work establishes that the choice of preference model is impactful, and our proposed regret preference model provides an improvement upon a core assumption of recent research. We have open sourced our experimental code, the human preferences dataset we gathered, and our training and preference elicitation interfaces for gathering a such a dataset.

## 1  Introduction

Improvements in reinforcement learning (RL) have led to notable recent achievements (Silver et al., 2016; Senior et al., 2020; Vinyals et al., 2019; Bellemare et al., 2020; Berner et al., 2019; Degrave et al., 2022; Wurman et al., 2022), increasing its applicability to real-world problems. Yet, like all optimization algorithms, even *perfect* RL optimization is limited by the objective it optimizes. For RL, this objective is created in large part by the reward function. Poor alignment between reward functions and the interests of human stakeholders limits the utility of RL and may even pose risks of financial cost and human injury or death (Amodei et al., 2016; Knox et al., 2021).

Influential recent research has focused on learning reward functions from preferences over pairs of trajectory segments, a common form of reinforcement learning from human feedback (RLHF). Nearly all of this recent work assumes that human preferences arise probabilistically from *only* the sum of rewards over a segment, i.e., the segment's **partial return** (Christiano et al., 2017; Sadigh et al., 2017; Ibarz et al., 2018; Bıyık et al., 2021; Lee et al., 2021a;b; Ziegler et al., 2019; Wang et al., 2022; Ouyang et al., 2022; Bai et al., 2022; Glaese et al., 2022; OpenAI, 2022). That is, these works assume that people tend to prefer trajectory segments that yield greater accumulated rewards *during the segment*. However, this preference model ignores seemingly important information about the segment's desirability, including the state values of the segment's start and end states. Separately, this partial return preference model can prefer suboptimal actions with lucky outcomes, like buying a lottery ticket.

This paper proposes an alternative preference model based on the **regret** of each segment, which is a measure of how much each segment deviates from optimal decision-making. More precisely, regret is the negated sum of an optimal policy's advantage of each transition in the segment (Section 2.2). Figures 1 and 2 show intuitive examples of when these two models disagree. Some examples of domains where the preference models will differ are those with constant reward until the end, including competitive games like chess, go, and soccer as well as tasks for which the objective is to minimize time until reaching a goal.

For these two preference models, we first focus theoretically on a normative analysis (Section 3)—i.e., what preference model would we *want* humans to use if we could choose one based on how informative its generated preferences are—proving that reward learning on infinite, exhaustive preferences with our proposed regret preference model identifies a reward function with the same set of optimal policies as the reward function with which the preferences are generated. We also prove that the partial return preference model is not guaranteed to identify such a reward function in three different contexts: without preference noise, when trajectories of different lengths are possible from a state, and when segments consist of only one transition. We follow up with a descriptive

analysis of how well each of these proposed models align with *actual* human preferences by collecting a human-labeled dataset of preferences in a rich grid world domain (Section 4) and showing that the regret preference model better predicts these human preferences (Section 5). Finally, we find that the policies ultimately created through the regret preference model tend to outperform those from the partial return model learning—both when assessed with collected human preferences or when assessed with synthetic preferences (Section 6). Our code for learning and for re-running our main experiments can be found here, alongside our interface for training subjects and for preference elicitation. The human preferences dataset is available here Knox et al. (2023).

In summary, our primary contributions are five-fold:

1. We propose a new model for human preferences that is based on regret instead of partial return.

2. We theoretically validate that this regret-based model has the desirable characteristic of reward identifiability, and that the partial return model does not.

3. We empirically validate that when each preference model learns from a preferences dataset it created, this regret-based model leads to better-aligned policies.

4. We empirically validate that, with a collected dataset of human preferences, this regret-based model both better describes the human preferences and leads to better-aligned policies.

5. Overall, we show that the choice of preference model impacts the alignment of learned reward functions.

## 2 Preference models for learning reward functions

We assume that the task environment is a Markov decision process (MDP) specified by the tuple ($S$, $A$, $T$, $\gamma$, $D_0$, $r$). $S$ and $A$ are the sets of possible states and actions, respectively. $T$ is a transition function, $T : S \times A \to p(\cdot|s,a)$; $\gamma$ is the discount factor; and $D_0$ is the distribution of start states. Unless otherwise stated, we assume all tasks are undiscounted (i.e., $\gamma = 1$) and have terminal states, after which only 0 reward can be received. Discounting is considered in depth in Appendix B.2. $r$ is a reward function, $r : S \times A \times S \to \mathbb{R}$, where the reward $r_t$ at time $t$ is a function of $s_t$, $a_t$, and $s_{t+1}$. An MDP\$r$ is an MDP without a reward function.

Throughout this paper, $r$ refers to the ground-truth reward function for some MDP; $\hat{r}$ refers to a learned approximation of $r$; and $\tilde{r}$ refers to any reward function (including $r$ or $\hat{r}$). A policy ($\pi : S \times A \to [0,1]$) specifies the probability of an action given a state. $Q_{\tilde{r}}^{\pi}$ and $V_{\tilde{r}}^{\pi}$ refer respectively to the state-action value function and state value function for a policy, $\pi$, under $\tilde{r}$, and are defined as follows.

$$V_{\tilde{r}}^{\pi}(s) \stackrel{\text{def}}{=} \mathbb{E}_{\pi}[\sum_{t=0}^{\infty} \tilde{r}(s_t,a_t,s_{t+1})|s_0 = s]$$

$$Q_{\tilde{r}}^{\pi}(s,a) \stackrel{\text{def}}{=} \mathbb{E}_{\pi}[\tilde{r}(s,a,s') + V_{\tilde{r}}^{\pi}(s')]$$

An optimal policy $\pi^*$ is any policy where $V_{\tilde{r}}^{\pi^*}(s) \geq V_{\tilde{r}}^{\pi}(s)$ at every state $s$ for every policy $\pi$. We write shorthand for $Q_{\tilde{r}}^{\pi^*}$ and $V_{\tilde{r}}^{\pi^*}$ as $Q_{\tilde{r}}^*$ and $V_{\tilde{r}}^*$, respectively. The optimal advantage function is defined as $A_{\tilde{r}}^*(s,a) \triangleq Q_{\tilde{r}}^*(s,a) - V_{\tilde{r}}^*(s)$; this measures how much an action reduces expected return relative to following an optimal policy.

Throughout this paper, the ground-truth reward function $r$ is used to algorithmically generate preferences when they are not human-generated, is hidden during reward learning, and is used to evaluate the performance of optimal policies under a learned $\hat{r}$.

### 2.1 Reward learning from pairwise preferences

A reward function can be learned by minimizing the cross-entropy loss—i.e., maximizing the likelihood—of observed human preferences, a common approach in recent literature (Christiano et al., 2017; Ibarz et al., 2018; Wang et al., 2022; Bıyık et al., 2021; Sadigh et al., 2017; Lee et al., 2021a;b; Ziegler et al., 2019; Ouyang et al., 2022; Bai et al., 2022; Glaese et al., 2022; OpenAI, 2022).

**Segments**     Let $\sigma$ denote a segment starting at state $s_0^\sigma$. Its length $|\sigma|$ is the number of transitions within the segment. A segment includes $|\sigma|+1$ states and $|\sigma|$ actions: $(s_0^\sigma, a_0^\sigma, s_1^\sigma, a_1^\sigma, ..., s_{|\sigma|}^\sigma)$. In this problem setting, segments lack any reward information. As shorthand, we define $\sigma_t \triangleq (s_t^\sigma, a_t^\sigma, s_{t+1}^\sigma)$. A segment $\sigma$ is **optimal** with respect to $\tilde{r}$ if, for every $i \in \{1, ..., |\sigma|\text{-}1\}$, $Q_{\tilde{r}}^*(s_i^\sigma, a_i^\sigma) = V_{\tilde{r}}^*(s_i^\sigma)$. A segment that is not optimal is **suboptimal**. Given some $\tilde{r}$ and a segment $\sigma$, $\tilde{r}_t^\sigma \triangleq \tilde{r}(s_t^\sigma, a_t^\sigma, s_{t+1}^\sigma)$, and the undiscounted **partial return** of a segment $\sigma$ is $\sum_{t=0}^{|\sigma|-1} \tilde{r}_t^\sigma$, denoted in shorthand as $\Sigma_\sigma \tilde{r}$.

**Preference datasets**     Each preference over a pair of segments creates a sample $(\sigma_1, \sigma_2, \mu)$ in a preference dataset $D_\succ$. Vector $\mu = \langle \mu_1, \mu_2 \rangle$ represents the preference; specifically, if $\sigma_1$ is preferred over $\sigma_2$, denoted $\sigma_1 \succ \sigma_2$, $\mu = \langle 1, 0 \rangle$. $\mu$ is $\langle 0, 1 \rangle$ if $\sigma_1 \prec \sigma_2$ and is $\langle 0.5, 0.5 \rangle$ for $\sigma_1 \sim \sigma_2$ (no preference). For a sample $(\sigma_1, \sigma_2, \mu)$, we assume that the two segments have equal lengths (i.e., $|\sigma_1| = |\sigma_2|$).

**Loss function**     To learn a reward function from a preference dataset, $D_\succ$, a common assumption is that these preferences were generated by a preference model $P$ that arises from an unobservable *ground-truth* reward function $r$. We approximate $r$ by minimizing cross-entropy loss to learn $\hat{r}$:

$$loss(\hat{r}, D_\succ) = -\sum_{(\sigma_1, \sigma_2, \mu) \in D_\succ} \mu_1 \log P(\sigma_1 \succ \sigma_2 | \hat{r}) + \mu_2 \log P(\sigma_1 \prec \sigma_2 | \hat{r}) \tag{1}$$

For a single sample where $\sigma_1 \succ \sigma_2$, the sample's likelihood is $P(\sigma_1 \succ \sigma_2 | \hat{r})$ and its loss is therefore $-\log P(\sigma_1 \succ \sigma_2 | \hat{r})$. If $\sigma_1 \prec \sigma_2$, its likelihood is $1 - P(\sigma_1 \succ \sigma_2 | \hat{r})$. This loss is under-specified until $P(\sigma_1 \succ \sigma_2 | \hat{r})$ is defined, which is the focus of this paper. We show that the common partial return model of preference probabilities is flawed and introduce an improved regret-based preference model.

**Preference models**     A preference model determines the probability of one trajectory segment being preferred over another, $P(\sigma_1 \succ \sigma_2 | \tilde{r})$. $P(\sigma_1 \succ \sigma_2 | \tilde{r}) + P(\sigma_1 \sim \sigma_2 | \tilde{r}) + P(\sigma_1 \prec \sigma_2 | \tilde{r}) = 1$, and $P(\sigma_1 \sim \sigma_2 | \tilde{r}) = 0$ for the preference models considered herein. Preference models could be applied to model preferences provided by humans or other systems. Preference models can also directly generate preferences, and in such cases we refer to them as **preference generators**.

## 2.2   Choice of preference model: partial return and regret

**Partial return**     All aforementioned recent work assumes human preferences are generated by a Boltzmann distribution over the two segments' partial returns, expressed here as a logistic function.[1]

$$P_{\Sigma r}(\sigma_1 \succ \sigma_2 | \tilde{r}) = logistic\left(\Sigma_{\sigma_1} \tilde{r} - \Sigma_{\sigma_2} \tilde{r}\right). \tag{2}$$

**Regret**     We introduce an alternative preference model based on the regret of each transition in a segment. We first focus on segments with deterministic transitions. For a transition $(s_t, a_t, s_{t+1})$ in a deterministic segment, $regret_\mathrm{d}(\sigma_t | \tilde{r}) \triangleq V_{\tilde{r}}^*(s_t^\sigma) - [\tilde{r}_t + V_{\tilde{r}}^*(s_{t+1}^\sigma)]$. The subscript $d$ in $regret_d$ signifies the assumption of deterministic transitions. For a full deterministic segment,

$$regret_d(\sigma | \tilde{r}) \triangleq \sum_{t=0}^{|\sigma|-1} regret_d(\sigma_t | \tilde{r}) = V_{\tilde{r}}^*(s_0^\sigma) - (\Sigma_\sigma \tilde{r} + V_{\tilde{r}}^*(s_{|\sigma|}^\sigma)), \tag{3}$$

with the right-hand expression arising from cancelling out intermediate state values. Therefore, deterministic regret measures how much the segment reduces expected return from $V_{\tilde{r}}^*(s_0^\sigma)$. An optimal segment, $\sigma^*$, always has 0 regret, and a suboptimal segment, $\sigma^{\neg *}$, will always have positive regret, an intuitively appealing property that also plays a role in the identifiability proof of Theorem 3.1.

---

[1] See Appendix B.1 for a derivation of this logistic expression from a Boltzmann distribution with a temperature of 1. Unless otherwise stated, we ignore the temperature because scaling reward has the same effect when preference probabilities are not deterministic. The temperature is allowed to vary for our theory in Section 3. Another context when the temperature parameter would be useful is when learning a single reward function with a loss function that includes one or more loss terms in addition to the formula in Equation 1; in such a case, scaling reward might undesirably affect the other loss term(s), whereas the varying the Boltzmann temperature changes the preference entropy without affecting the other loss term(s).

Stochastic state transitions, however, can result in $regret_d(\sigma^*|\hat{r}) > regret_d(\sigma^{\neg *}|\tilde{r})$, losing the property above. For instance, an optimal action can lead to worse return than a suboptimal action, based on stochasticity in state transitions. To retain this property that optimal segments have a regret of 0 and suboptimal segments have positive regret, we first note that the effect on expected return of transition stochasticity from a transition $(s_t, a_t, s_{t+1})$ is $[\tilde{r}_t + V_{\tilde{r}}^*(s_{t+1})] - Q_{\tilde{r}}^*(s_t, a_t)$ and add this expression once per transition to get $regret(\sigma)$, removing the subscript $d$ that refers to determinism. The regret for a single transition becomes $regret(\sigma_t|\tilde{r}) = [V_{\tilde{r}}^*(s_t^\sigma) - [\tilde{r}_t + V_{\tilde{r}}^*(s_{t+1}^\sigma)]] + [[\tilde{r}_t + V_{\tilde{r}}^*(s_{t+1}^\sigma)] - Q_{\tilde{r}}^*(s_t^\sigma, a_t^\sigma)] = V_{\tilde{r}}^*(s_t^\sigma) - Q_{\tilde{r}}^*(s_t^\sigma, a_t^\sigma) = -A_{\tilde{r}}^*(s_t^\sigma, a_t^\sigma)$. Regret for a full segment is

$$regret(\sigma|\tilde{r}) = \sum_{t=0}^{|\sigma|-1} regret(\sigma_t|\tilde{r}) = \sum_{t=0}^{|\sigma|-1} \left[ V_{\tilde{r}}^*(s_t^\sigma) - Q_{\tilde{r}}^*(s_t^\sigma, a_t^\sigma) \right] = \sum_{t=0}^{|\sigma|-1} -A_{\tilde{r}}^*(s_t^\sigma, a_t^\sigma). \tag{4}$$

The regret preference model is the Boltzmann distribution over negated regret:

$$P_{regret}(\sigma_1 \succ \sigma_2|\tilde{r}) \triangleq logistic\Big(regret(\sigma_2|\tilde{r}) - regret(\sigma_1|\tilde{r})\Big). \tag{5}$$

Lastly, we note that if two segments have deterministic transitions, end in terminal states, and have the same starting state, in this special case the regret model reduces to the partial return model: $P_{regret}(\cdot|\tilde{r}) = P_{\Sigma r}(\cdot|\tilde{r})$.

In this article, our *normative* results examine both tasks with deterministic transitions and tasks with stochastic transitions. These normative results include the theoretical analysis in Section 3 and the empirical results with synthetic data in Section 6.2 and Appendix F.2, with stochastic tasks specifically examined empirically in Appendix F.2.4. We gather human preferences for a deterministic task, which allows us to investigate the results with the more intuitive expression of $regret_d$ that includes partial return as a component.

**Algorithms in this paper** All algorithms in the body of this paper can be summarized as "minimize Equation 1". They differ only in how the preference probabilities are calculated. All reward function learning via partial return uses Equation 2, replicating the dominant algorithm in recent literature (Christiano et al., 2017; Ibarz et al., 2018; Wang et al., 2022; Bıyık et al., 2021; Sadigh et al., 2017; Lee et al., 2021a;b; Ouyang et al., 2022). We use two algorithms for reward function learning via regret. The theory in Section 3 assumes exact measurement of regret, using Equation 5. Section 6 introduces Equation 6 to approximate regret—replacing Equation 5 to create another algorithm—and uses the resulting algorithm for our experimental results later in that section. Appendix B introduces other algorithms that use Equation 1, as well as one in Appendix B.4 that generalizes Equation 1.

**Regret as a model for human preference** $P_{regret}$ makes at least three assumptions worth noting. First, it keeps the assumption that human preferences follow a Boltzmann distribution over some statistic, which is a common model of choice behavior in economics and psychology, where it is called the Luce-Shepard choice rule (Luce, 1959; Shepard, 1957). Second, $P_{regret}$ implicitly assumes humans can identify optimal and suboptimal segments when they see them, which will be less true in domains where the human has less expertise. This assumption is similar to a common assumption of many algorithms for imitation learning, that humans can provide demonstrations that are optimal or noisily optimal (e.g., Abbeel & Ng (2004)).

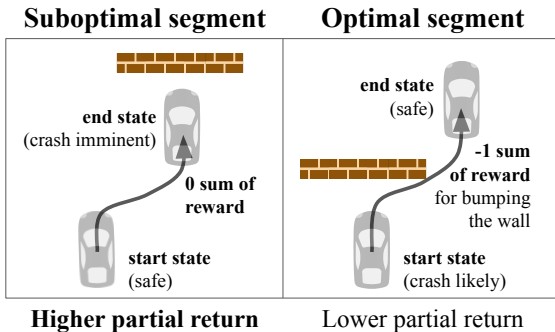

| Suboptimal segment | Optimal segment |
| --- | --- |

**Higher partial return**
Higher regret

Lower partial return
**Lower regret**

Figure 2: Two segments of a car moving at high speed near a brick wall. On the left, a car moves toward a brick wall; a bad crash is imminent, but has not yet occurred. On the right, a car escapes an imminent crash against a brick wall with only a scrape. Assume the right segment is optimal and the left segment is suboptimal (as defined in Sec. 2.1). The left segment has a higher sum of reward, so it is preferred under the partial return preference model. The right segment is preferred under the regret preference model since optimal segments have minimal regret. If we also assume deterministic transitions, then the regret model includes the difference in values between the start state and the end state (Equation 3), and the right segment would tend to be preferred because it greatly improves its state values from start to end, whereas the left segment's state values greatly worsen. We suspect our human readers will also tend to prefer the right segment.

Lastly, $P_{regret}$ assumes that in stochastic settings where the best *outcome* may only result from suboptimal decisions (e.g., buying a lottery ticket), humans instead

prefer optimal *decisions.* We suspect humans are capable of expressing either type of preference—based on decision quality or desirability of outcomes—and can be influenced by training or the preference elicitation interface. Curiously, for stochastic tasks in which preferences are based upon segments' observed outcomes, a preference model that uses deterministic $regret_d$ in Equation 5 appears fitting, since it does not subtract out the effects of fortunate and unfortunate transitions but does include segments' start and end state values.

In practice we determine that the regret model produces improvements over the partial return model (Section 6), and its assumptions represent an opportunity for follow-up research.

## 3   Theoretical comparison

In this section, we consider how different ways of *generating preferences* affect reward inference, setting aside whether humans can be influenced to give preferences in accordance with a specific preference method. In economics, this analysis—and all of our later analyses with synthetic preferences—could be considered a normative analysis. In artificial intelligence, this analysis might be cast as a step towards defining criteria for a rational preference model.

The theorems and proofs below focus on identifiability, a property which determines whether the parameters of a model can be recovered from infinite, exhaustive samples generated by the model. A model is unidentifiable if any two parameterizations of the model result in the same model behavior. In our setting, the model of concern is a preference model and the parameters constitute the ground-truth reward function, $r$. A preference model is identifiable if an infinite, exhaustive preferences dataset created by the preference model contains sufficient information to infer a behaviorally equivalent reward function to $r$. Note that *identifiability focuses on the preference model alone as a preference generator*, not on the learning algorithm that uses such a preference model.

This section uses preference models that include discounting (see Appendix B.2). We allow for discounting to make the theory more general and also because discounting is integral to Section 3.2.3. Here the notation for $Q^*_{\tilde{r}}(s,a)$ and $V^*_{\tilde{r}}(s)$ is expanded to $Q^*_{(\tilde{r},\tilde{\gamma})}(s,a)$ and $V^*_{(\tilde{r},\tilde{\gamma})}(s)$ respectively include the discount factor. To make the other content in this section specific to undiscounted tasks, simply assume all instances of $\tilde{\gamma}=1$, including the ground-truth $\gamma$ and the $\hat{\gamma}$ used during reward function inference and policy improvement.

**Definition 3.1** (An identifiable preference model)**.** *For a preference model $P$, assume an infinite dataset $D_\succ$ of pairs of segments is constructed by repeatedly choosing $(\sigma_1,\sigma_2)$ and sampling a label $\mu \sim P(\sigma_1 \succ \sigma_2|r)$, using $P$ as a preference generator. Further assume that, in this dataset, all possible segment pairs appear infinitely many times. For some $M$ that is an MDP$\backslash(r,\gamma)$—an MDP with neither a reward function nor a discount factor—let $M_{(\tilde{r},\tilde{\gamma})}$ be $M$ with the reward function $\tilde{r}$ and the discount factor $\tilde{\gamma}$. Let $\Pi^*_{(\tilde{r},\tilde{\gamma})}$ be the set of optimal policies for $M_{(\tilde{r},\tilde{\gamma})}$. Let problem-equivalence class $\mathfrak{R}$ be the set of all pairs of a reward function and a discount factor such that if $(r_1,\gamma_1),(r_2,\gamma_2) \in \mathfrak{R}$ then $\Pi^*_{(r_1,\gamma_1)} = \Pi^*_{(r_2,\gamma_2)}$. Preference model $P$ is **identifiable** if and only if, for any choice of segment length $n$ and ground-truth $M_{(r,\gamma)}$, there exists an operation on $D_\succ$ that always outputs a $(\hat{r},\hat{\gamma})$ that is in the same problem equivalence class as $(r,\gamma)$. I.e., $\Pi^*_{(r,\gamma)} = \Pi^*_{(\hat{r},\hat{\gamma})}$.*

### 3.1   The regret preference model is identifiable.

We first prove that our proposed regret preference model is identifiable.

**Theorem 3.1** ($P_{regret}$ is identifiable)**.** *Let $P_{regret}$ be any function such that if $regret(\sigma_1|\tilde{r},\tilde{\gamma}) < regret(\sigma_2|\tilde{r},\tilde{\gamma})$, $P_{regret}(\sigma_1 \succ \sigma_2|\tilde{r},\tilde{\gamma}) > 0.5$, and if $regret(\sigma_1|\tilde{r},\tilde{\gamma}) = regret(\sigma_2|\tilde{r},\tilde{\gamma})$, $P_{regret}(\sigma_1 \succ \sigma_2|\tilde{r},\tilde{\gamma}) = 0.5$. $P_{regret}$ is identifiable.*

This class of regret preference models includes but is not limited to the Boltzmann distribution of Equation 5. Additionally, it includes a version of the regret preference model that noiselessly always prefers the segment with lower regret, as Theorem 3.2 considers for the partial return preference model.[2]

---

[2]Equations 2 and 5 can be extended to include such noiseless preference models by including the temperature parameter of the Boltzmann distributions (after converting from their logistic formulations, reversing the derivation in Appendix B.1), where we assume that setting the temperature to 0 results in a hard maximum. In other words, when the temperature is 0 the preference is given deterministically to the segment with the higher partial return in Equation 2 or regret in Equation 5.

Consider reviewing the definitions of optimal segments and suboptimal segments in Section 2.1 before proceeding.

For the proof below, we will apply the following **sufficiency test for identifiability**. Preference model $P$ is identifiable if, for any ground-truth $M_{(r,\gamma)}$, any $(\hat{r},\hat{\gamma}) = argmin_{(\tilde{r},\tilde{\gamma})}[loss(\tilde{r},\tilde{\gamma},D_{\succ})]$—for the cross-entropy loss (Eqn. 8, which is Eqn. 1 generalized to include discounting), with $P$ as the preference model—is in the same problem equivalence class as $(r,\gamma)$. I.e., $\Pi^*_{(r,\gamma)} = \Pi^*_{(\hat{r},\hat{\gamma})}$.

**Proof**     Make all assumptions in Definition 3.1. Let $(\hat{r},\hat{\gamma}) = argmin_{(\tilde{r},\tilde{\gamma})}[loss(\tilde{r},\tilde{\gamma},D_{\succ})]$, where loss is the cross-entropy loss from Eqn. 8 with $P_{regret}$ as the preference model.

Since $(\hat{r},\hat{\gamma})$ minimizes cross-entropy loss and is chosen from the complete space of reward functions and discount factors, $P_{regret}(\cdot|r,\gamma) = P_{regret}(\cdot|\hat{r},\hat{\gamma})$ for all possible segment pairs. Also, by Equation 12 (which generalizes Equation 4 to include discounting) $regret(\sigma|\tilde{r},\tilde{\gamma}) = 0$ if and only if $\sigma$ is optimal with respect to $\tilde{r}$. And $regret(\sigma|\tilde{r},\tilde{\gamma}) > 0$ if and only if $\sigma$ is suboptimal with respect to $(\tilde{r},\tilde{\gamma})$.

With respect to some $(\tilde{r}, \tilde{\gamma})$, let $\sigma^*$ be any optimal segment and $\sigma^{\neg*}$ be any suboptimal segment. $regret(\sigma^*|\tilde{r},\tilde{\gamma}) < regret(\sigma^{\neg*}|\tilde{r},\tilde{\gamma})$, so $P_{regret}(\sigma^* \succ \sigma^{\neg*}|\tilde{r},\tilde{\gamma}) > 0.5$. $P_{regret}(\cdot|\tilde{r},\tilde{\gamma})$ induces a total ordering over segments, defined by $regret(\sigma_1|\tilde{r},\tilde{\gamma}) < regret(\sigma_2|\tilde{r},\tilde{\gamma}) \iff P_{regret}(\sigma_1 \succ \sigma_2|\tilde{r},\tilde{\gamma}) > 0.5 \iff \sigma_1 > \sigma_2$ and $regret(\sigma_1|\tilde{r},\tilde{\gamma}) = regret(\sigma_2|\tilde{r},\tilde{\gamma}) \iff P_{regret}(\sigma_1 \succ \sigma_2|\tilde{r},\tilde{\gamma}) = 0.5 \iff \sigma_1 = \sigma_2$. Because regret has a minimum (0), there must be a set of segments which are ranked highest under this ordering, denoted $\Sigma^*_{(\tilde{r},\tilde{\gamma})}$. These segments in $\Sigma^*_{(\tilde{r},\tilde{\gamma})}$ are exactly those that achieve the minimum regret (0) and so are optimal with respect to $(\tilde{r},\tilde{\gamma})$.

Since the dataset ($D_{\succ}$) contains all segments by assumption, $\Sigma^*_{(\tilde{r},\tilde{\gamma})}$ contains all optimal segments with respect to $(\tilde{r},\tilde{\gamma})$. If a state-action pair $(s,a)$ is in an optimal segment, then by the definition of an optimal segment $Q^*_{(\tilde{r},\tilde{\gamma})}(s,a) = V^*_{(\tilde{r},\tilde{\gamma})}(s)$. The set of optimal policies $\Pi^*_{\tilde{r}}$ for $\tilde{r}$ is all $\pi$ such that, for all $(s,a)$, if $\pi(s,a) > 0$, then $Q^*_{(\tilde{r},\tilde{\gamma})}(s,a) = V^*_{(\tilde{r},\tilde{\gamma})}(s)$. In short, $\Sigma^*_{(\tilde{r},\tilde{\gamma})}$ determines the set of each state-action pair $(s,a)$ such that $Q^*_{(\tilde{r},\tilde{\gamma})}(s,a) = V^*_{(\tilde{r},\tilde{\gamma})}(s)$. This set determines $\Pi^*_{(\tilde{r},\tilde{\gamma})}$. Therefore $\Sigma^*_{(\tilde{r},\tilde{\gamma})}$ determines $\Pi^*_{(\tilde{r},\tilde{\gamma})}$, and we will refer to this determination as the function $g$.

We now focus on the reward function and discount factor used to generate preferences, $(r,\gamma)$, and on the inferred reward function and discount factor, $(\tilde{r},\tilde{\gamma})$. Since $P_{regret}(\cdot|r,\gamma) = P_{regret}(\cdot|\hat{r},\hat{\gamma})$, $(r,\gamma)$ and $(\hat{r},\hat{\gamma})$ induce the same total ordering over segments, and so $\Sigma^*_{(r,\gamma)} = \Sigma^*_{(\hat{r},\hat{\gamma})}$. Therefore $g(\Sigma^*_{(r,\gamma)}) = g(\Sigma^*_{(\hat{r},\hat{\gamma})})$. Since $g(\Sigma^*_{(r,\gamma)}) = \Pi^*_{(r,\gamma)}$ and $g(\Sigma^*_{(\hat{r},\hat{\gamma})}) = \Pi^*_{(\hat{r},\hat{\gamma})}$, $\Pi^*_{(r,\gamma)} = \Pi^*_{(\hat{r},\hat{\gamma})}$. $\qquad\square$

The proof above establishes the identifiability of $P_{regret}$ regardless of whether preferences are generated noiselessly or stochastically.

### 3.2   The partial return preference model is not generally identifiable.

In this subsection, we critique the previous standard preference model, the partial return model $P_{\Sigma r}$, by proving that this model can be unidentifiable in three different contexts.

- **Given *noiseless* preference labeling** by $P_{\Sigma r}$ in some MDPs, preferences never provide sufficient information to recover the set of optimal policies.

- **In variable-horizon tasks when the lengths of both segments in a pair are always equivalent.** Variable-horizon tasks include common tasks that terminate upon reaching success or failure states, reward functions that differ by a constant can have different sets of optimal policies. Yet for two such reward functions, the preference probabilities according to partial return will be identical.

- **With segment lengths of 1** ($|\sigma| = 1$), the discount factor $\gamma$ does not affect the partial return preference model and therefore will not be recoverable from the preferences it generates. Since different values of $\gamma$ can determine different sets of optimal policies, an inability to recover $\gamma$ is a third type of unidentifiability.

We now prove in each of these three contexts that the partial return preference model is not identifiable.

For each, we will apply the following **sufficiency test for non-identifiability**. Preference model $P$ is *not* identifiable if there exist two ground-truth MDPs, $M_{(r_1,\gamma_1)}$ and $M_{(r_2,\gamma_2)}$, such that $\Pi^*_{(r_1,\gamma_1)} \neq \Pi^*_{(r_2,\gamma_2)}$ and the infinite preference datasets created as described in Definition 3.1 by $P$ for $M_{(r_1,\gamma_1)}$ and $M_{(r_2,\gamma_2)}$ are identical. Note that such identical preference datasets lack the information to differentiate which MDP they came from.

### 3.2.1 Partial return is not identifiable when preferences are noiseless.

**Theorem 3.2** (Noiseless $P_{\Sigma r}$ is not identifiable)**.** *Let $P_{\Sigma r}$ be any function such that if $\Sigma_{\sigma_1}\tilde{r} > \Sigma_{\sigma_2}\tilde{r}$, $P_{\Sigma r}(\sigma_1 \succ \sigma_2|\tilde{r}) = 1$ and if $\Sigma_{\sigma_1}\tilde{r} = \Sigma_{\sigma_2}\tilde{r}$, $P_{\Sigma r}(\sigma_1 \succ \sigma_2|\tilde{r}) = 0.5$. There exists an MDP in which $P_{\Sigma r}$ is not identifiable.*

Below we present two proofs of Theorem 3.2. Each are proofs by counterexample. The first is a general proof and the second proof assumes a common characteristic, that all segments in the preference dataset are the same length. Though only one proof is needed, we present two because each counterexample demonstrates a qualitatively different category of how the partial return preference model can fail to identify the set of optimal policies.

**Proof based on stochastic transitions:** Assume the following class of MDPs, illustrated in Figure 3. The agent always begins at start state $s_0$. From $s_0$, action $a_{safe}$ always transitions to $s_{safe}$, getting a reward of 0. From $s_0$, action $a_{risk}$ transitions to $s_{win}$ with probability 0.5, getting a reward of $r_{win}$, and transitions to $s_{lose}$ with with probability 0.5, getting a reward of $-10$. In all MDPs in this class, $r_{win} > 0$. All 3 possible resulting states ($s_{safe}$, $s_{win}$, and $s_{lose}$) are absorbing states, from which all further reward is 0.

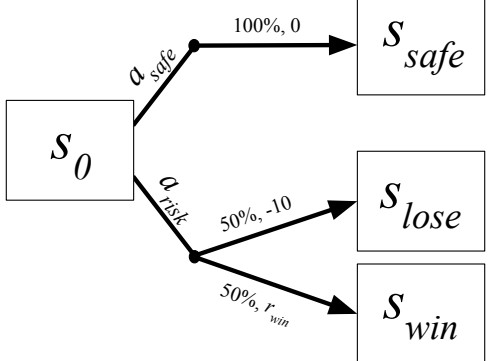

Figure 3: A class of MDPs in which, if $r_{win} > 0$, the partial return preference model fails the test for identifiability.

If $r_{win} \geq 10$, $a_{risk}$ is optimal in $s_0$. If $r_{win} \leq 10$, $a_{safe}$ is optimal in $s_0$. Three single-transition segments exist: $(s_0, a_{safe}, s_{safe})$, $(s_0, a_{risk}, s_{win})$, and $(s_0, a_{risk}, s_{lose})$. By noiseless $P_{\Sigma r}$, $(s_0, a_{risk}, s_{win}) \succ (s_0, a_{safe}, s_{safe}) \succ (s_0, a_{risk}, s_{lose})$, regardless of the value of $r_{win}$. In other words, $P_{\Sigma r}$ is insensitive to what the optimal action is from $s_0$ in this class of MDPs.

Now assume MDP $M$, where $r_{win} = 11$. In linear form, the weight vector for the reward function $r_M$ can be expressed as $w_{r_{M_1}} = <-10, 0, 11>$. Let $\hat{r}_M$ have $w_{\hat{r}_M} = <-10, 0, 9>$. Both $r_M$ and $\hat{r}_M$ have the same preferences as above, meaning that $\hat{r}_M$ minimizes loss on an infinite preferences dataset $D_\succ$ created by $P_{\Sigma r}$, yet it has a different optimal policy. Therefore, noiseless $P_{\Sigma r}$ is not identifiable. $\qquad\square$

In contrast, note that by noiseless $P_{regret}$, the preferences are different than those above for $P_{\Sigma r}$. If $r_{win} > 10$, then $(s_0, a_{risk}, s_{win}) \sim (s_0, a_{risk}, s_{lose}) \succ (s_0, a_{safe}, s_{safe})$, If $r_{win} < 10$, then $(s_0, a_{safe}, s_{safe}) \succ (s_0, a_{risk}, s_{win}) \sim (s_0, a_{risk}, s_{lose})$. Intuitively, this difference comes from $P_{regret}$ always giving higher preference probability to optimal actions, even if they result in bad outcomes. Another perspective can be found from the utility theory of Von Neumann & Morgenstern (1944). Specifically, $P_{\Sigma r}$ gives preferences over outcomes, which in the terms of utility theory can only be used to infer an ordinal utility function. Ordinal utility functions are merely consistent with the preference ordering over outcomes and do not generally capture preferences over actions when their outcomes are stochastically determined. The deterministic regret preference model, $P_{regret_d}$, also has this weakness in tasks with stochastic transitions. On the other hand, $P_{regret}$ forms preferences over so-called lotteries—the distribution over possible outcomes—and can therefore learn a cardinal utility function, which can explain preferences over risky action. See Russell & Norvig (2020, Ch. 16) for more detail on these concepts from expected utility theory.

Since the proof above focuses upon stochastic transitions, we show the lack of identifiability for noiseless $P_{\Sigma r}$ can be found for quite different reasons in a deterministic MDP when the preferences dataset has a common characteristic: that all segments are the same length.

**Proof based on segments of fixed length:** Consider the MDP $M_1$ in Figure 4 and assume preferences are given over segments with length 1 (i.e., containing one transition). The optimal policy for $M_1$ is to move rightward from $s_0$, whereas optimal behavior for $M_1'$ is to move downward from $s_0$. In *both* $M_1$ and $M_1'$, preferences by $P_{\Sigma r}$ are as follows, omitting the action for brevity: $(s_a,s_0) \sim (s_a,s_{term}) \sim (s_0,s_a) \succ (s_0,s_{term})$. As in the previous proof, $P_{\Sigma r}$ is insensitive to certain changes in the reward function that alter the set of optimal policies. Whenever this characteristic is found, $\Pi_r^* = \Pi_{\hat{r}}^*$ is not guaranteed, failing the test for identifiability. Here specifically, the reward function for $M_1'$ would achieve the minimum possible cross-entropy loss on an exhaustive preference dataset created in $M_1$ with the noiseless preferences from the partial return preference model, despite the optimal policy in $M_1'$ conflicting with the ground-truth optimal policy.

The logic of this proof can be applied for trajectories of length 2 in the MDP $M_2$ shown in Figure 5. Together, $M_1$ and $M_2$ suggest a rule for constructing an MDP where $\Pi_r^* = \Pi_{\hat{r}}^*$ is not guaranteed for $P_{\Sigma r}$, failing the identifiability test for any fixed segment length, $|\sigma|$: set the number of states to the right of $s_0$ to $|\sigma|$ (not counting $s_{term}$), set the reward $r_{fail}$ for $(s_0, s_{term})$ such that $r_{fail} < 0$, and set the reward for each other transition to $c + r_{fail}/(|\sigma|+1)$, where $c > 0$. Given an MDP constructed this way, an alternative reward function that results in the same preferences under $P_{\Sigma r}$ yet has a different optimal action from $s_0$ can then be constructed by

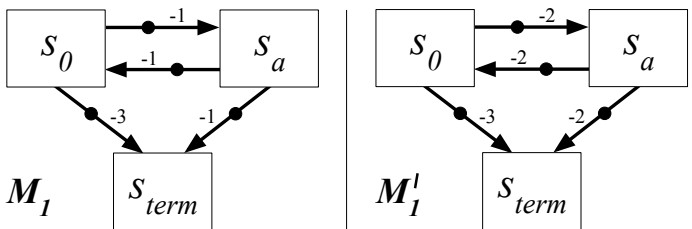

Figure 4: An MDP ($M_1$) where $\Pi_r^* = \Pi_{\hat{r}}^*$ is not guaranteed for the partial return preference model, failing the test for identifiability with segments of length 1. The ground-truth reward function is shown to the left, and an MDP $M_1'$ with an alternative reward function is shown to the right. Under partial return, both create the same set of preferences despite having different optimal actions from $s_0$.

changing all reward other than $r_{fail}$ to $c + r_{fail}/(|\sigma|+1)$, where $c$ now is constrained to $c < 0$ and $c \times |\sigma| < r_{fail}$. Note that the set of preferences for each of these MDPs is the same even when including segments that reach terminal state before $|\sigma|$ transitions (which can still be considered to be of length $|\sigma|$ if the terminal state is an absorbing state from which reward is 0).

**Discussion of preference noise and identifiability** Of the two proofs by example for Theorem 3.2, the first proof's example reveals issues when learning reward functions with stochastic transitions with either $P_{\Sigma r}$ or *deterministic* $P_{regret_d}$. These issues directly correspond to the need for preferences over distributions over outcomes (i.e., lotteries) to construct a cardinal utility function (see Russell & Norvig (2020, Ch. 16)). Correspondingly, when Skalse et al. (2022) consider reward identifiability with the partial return preference model, they change the learning problem such that a training sample consists of preferences between *distributions* over trajectories. Intuitively, Theorem 3.2 says that $P_{\Sigma r}$ is not identifiable without the distribution over preferences providing information about the proportions of rewards with respect to each other. In contrast, to be identifiable, the regret preference model does not require this preference error (though it can presumably benefit from it in certain contexts).

### 3.2.2 Partial return is not identifiable in variable-horizon tasks.

Many common tasks have the characteristic of having at least one state from which trajectories of *different*

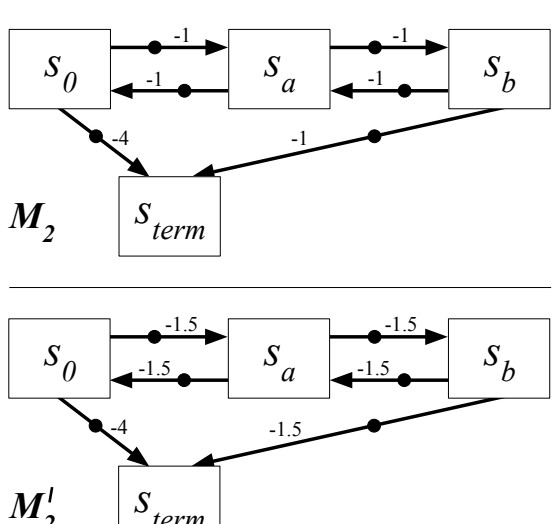

Figure 5: An MDP ($M_2$) where $\Pi_r^* = \Pi_{\hat{r}}^*$ is not guaranteed for the partial return preference model, failing the test for identifiability with segments of length 2. The ground-truth reward function is shown in the top diagram, and an MDP $M_2'$ with an alternative reward function is shown in the bottom diagram. Under partial return, both create the same set of preferences despite having different optimal actions from $s_0$.

lengths are possible, which we refer to as being a **variable-horizon task**. Tasks that terminate upon completing a goal typically have this characteristic. We also assume that for any pair of segments in the preference dataset, the lengths of those two segments are equivalent. This assumption describes typical practice (e.g., Christiano et al. (2017); Sadigh et al. (2017); Ibarz et al. (2018); Bıyık et al. (2021); Lee et al. (2021a); Wang et al. (2022)). In this context, we show another way that the partial return preference model is not identifiable, a limitation that has arisen dramatically in our own experiments and is not limited to noiseless preferences: *adding a constant to a reward function will often change the set of optimal policies, but it will not change the probability of preference for any two segments. Therefore, those preferences will not contain information to recover the set of optimal policies.*

We now explain why such a constant shift will not change the probability of preference based upon partial return. Consider a constant value $c$ and two reward functions, $r_1$ and $r_2$, where $r_1(s_t,a_t,s_{t+1}) - r_2(s_t,a_t,s_{t+1}) = c$ for all transitions $(s_t,a_t,s_{t+1})$. The partial return of any segment of length $|\sigma|$ will be $c \times |\sigma|$ higher for $r_1$ than for $r_2$ (assuming an undiscounted task, $\gamma = 1$). In the partial return preference model (Equation 2), this addition of $c \times |\sigma|$ to each segment's partial return cancels out, having no effect on the different in the segments' partial returns and therefore also having no effect on the preference probabilities. Consequently, adding $c$ to a reward function's output will also not affect the distribution over preference datasets that the partial return preference model would create via sampling from its preference probabilities.

If, for each state in an MDP, all possible trajectories from that state have the *same* length, then adding a constant $c$ to the output of the reward function does not affect the set of optimal policies. Specifically, the set of optimal policies is preserved because the return for any trajectory from a state is changed by $c \times |\tau|$, where $|\tau|$ is the unchanging trajectory length from that state, so the ranking of trajectories by their returns is unchanged and also the expected return of a policy from that state is unchanged. Continuing tasks and fixed-horizon tasks have this property.

However, if trajectories from a state can terminate after *different* numbers of transitions, then two reward functions that differ by a constant can have different sets of optimal policies. Episodic tasks are often vulnerable to this issue. To illustrate, consider the task in Figure 1, a simple grid world task that penalizes the agent with $-1$ reward for each step it takes to reach the goal. If this reward per step is shifted to $+1$ (or any positive value), then any optimal policy will *avoid the goal*, flipping the objective of the task from that of reaching the goal. So, for variable-horizon tasks, $P_{\Sigma r}$ is not generally identifiable.

Though identifiability focuses on what information is encoded in preferences, this issue has practical consequences during *learning* from preferences over segments of length 1: for a preferences dataset, all reward functions that differ by a constant assign the same likelihood to the dataset, making the choice between such reward functions arbitrary and the learning problem underspecified. Some past authors have acknowledged this insensitivity to a shift (Christiano et al., 2017; Lee et al., 2021a; Ouyang et al., 2022; Hejna & Sadigh, 2023), and the common practice of forcing all tasks to have a fixed horizon (Christiano et al., 2017; Gleave et al., 2022) may be partially attributable to $P_{\Sigma r}$'s lack of identifiability in variable-horizon tasks, leading to its low performance in such tasks. In Appendix F.2.2, we propose a stopgap solution to this problem and also observe that in episodic grid worlds that the partial return preference model performs catastrophically poorly without this solution, both with synthetic preferences and human preferences.

**The regret preference model is appropriately affected by constant reward shifts.** Here we give intuition for why adding a constant $c$ to the output of a reward function does not threaten the identifiability of the regret preference model, as established in Theorem 3.1. As stated above, adding $c$ to reward can change the set of optimal policies. Any such change in what actions are optimal would likewise change the ordering of segments by regret, so the likelihood of a preferences dataset according to the regret preference model *would* be affected by such a constant shift in the learned reward function (as it should be).

### 3.2.3   Partial return is not identifiable for segment lengths of 1.

Arguably the most impactful application to date of learning reward functions from human preferences is to fine-tune large language models. For the most notable of these applications, the segment length $|\sigma| = 1$ (Ziegler et al., 2019; OpenAI, 2022; Glaese et al., 2022; Ouyang et al., 2022; Bai et al., 2022).

Changing $\gamma$ often changes the set of optimal policies, yet when $|\sigma|=1$, changing the discount factor does not change preference probabilities based upon partial return preference model. We elaborate below.

Here we make an exception to this article's default assumption that all tasks are undiscounted. As we describe in Appendix B.2, the discounted partial return of a segment is $\sum_{t=0}^{|\sigma|-1} \tilde{\gamma}^t \tilde{r}_t^\sigma$. We follow the standard convention that $0^0=1$. When $|\sigma|=1$, the partial return simplifies to the immediate reward, $\tilde{r}_0^\sigma$, regardless of the value of $\gamma$. Consequently the partial return preference model is unaffected by the discount factor when $|\sigma|=1$. We leave to the reader the task of a precise proof by counterexample that partial return is not identifiable when $|\sigma|=1$; any two MDPs that differ only by their discount factor and have different sets of optimal policies will suffice to provide a proof, since the distribution of preferences according to partial return will be identical in each of these MDPs, establishing the lack of identifiability.

To remove this source of unidentifiability, the preferences dataset would need to be presented to the learning algorithm with a corresponding discount factor. Past work on identifiability in this setting (Skalse et al., 2022) has assumed that the discount factor is given and does not discuss the topic further.

As with the other identifiability issues demonstrated in this subsection, this issue has practical consequences *during learning* from preferences. When $|\sigma|=1$, the choice of $\hat{\gamma}$ is arbitrary, making the learning problem underspecified.

**The regret preference model is identifiable even when the discount factor is unknown.** Note that Theorem 3.1 already includes this case. To add some intuition, the discounted regret of a segment—presented in Appendix B.2—does include the discount factor in its formulation, regardless of segment length. Therefore, the discount factor used during preference generation does impact what reward function is learned.

## 4 Creating a human-labeled preference dataset

To empirically investigate the consequences of each preference model when learning reward from *human* preferences, we collected a preference dataset labeled by human subjects via Amazon Mechanical Turk. This data collection was IRB-approved. Appendix D adds detail to the content below.

### 4.1 The general delivery domain

The delivery domain consists of a grid of cells, each of a specific road surface type. The delivery agent's state is its location. The agent's action space is moving one cell in one of the four cardinal directions. The episode can terminate either at the destination for $+50$ reward or in failure at a sheep for $-50$ reward. The reward for a non-terminal transition is the sum of any reward components. Cells with a white road surface have a $-1$ reward component, and cells with brick surface have a $-2$ component. Additionally, each cell may contain a coin $(+1)$ or a roadblock $(-1)$. Coins do not disappear and at best cancel out the road surface cost. Actions that would move the agent into a house or beyond the grid's perimeter result in no motion and receive reward that includes the current cell's surface reward component but not any coin or roadblock components. In this work, the start state distribution, $D_0$, is always uniformly random over non-terminal states. This domain was designed to permit subjects to easily identify bad behavior yet also to be difficult for them to determine *optimal* behavior from most states, which is representative of many common tasks. Note

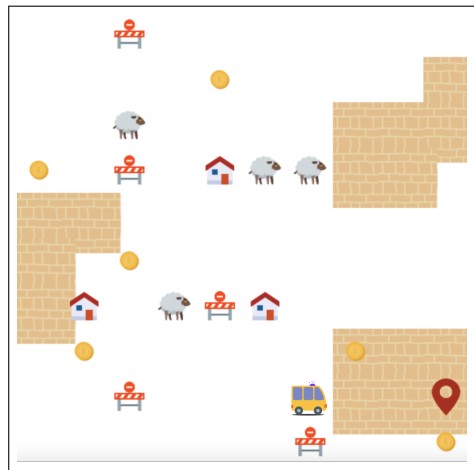

Figure 6: The delivery task used to gather human preferences. The yellow van is the agent and the red inverted teardrop is the destination.

that this intended difficulty forces some violation of the regret preference model's assumption that humans always prefer optimal segments over suboptimal ones, therefore testing its performance in non-ideal conditions.

### 4.1.1 The delivery task

We chose one instantiation of the delivery domain for gathering our dataset of human preferences. This specific MDP has a $10\times10$ grid. From every state, the highest return possible involves reaching the goal, rather than hitting a sheep or perpetually avoiding termination. Figure 6 shows this task.

## 4.2 The subject interface and survey

This subsection describes the three main stages of each data collection session. A video showing the full experimental protocol can be seen at youtu.be/V3hAqlE0qXg.

**Teaching subjects about the task** Subjects first view instructions describing the general domain. To avoid the jargon of "return" and "reward," these terms are mapped to equivalent values in US dollars, and the instructions describe the goal of the task as maximizing the delivery vehicle's financial outcome, where the reward components are specific financial impacts. This information is shared amongst interspersed interactive episodes, in which the subject controls the agent in domain maps that are each designed to teach one or two concepts. Our intention during this stage is to inform the later preferences of the subject by teaching them about the domain's dynamics and its reward function, as well as to develop the subject's sense of how desirable various behaviors are. At the end of this stage, the subject controls the agent for two episodes in the specific delivery task shown in Figure 6.

**Preference elicitation** After each subject is trained to understand the task, they indicate their preferences between 40–50 randomly-ordered pairs of segments, using the interface shown in Figure 7. The subjects select a preference, no preference ("same"), or "can't tell". In this work, we exclude responses labeled "can't tell", though one might alternatively try to extract information from these responses.

**Subjects' task comprehension** Subjects then answered questions testing their understanding of the task, and we removed their data if they scored poorly. We also removed a subject's data if they preferred colliding the vehicle into a sheep over not doing so, which we interpreted as poor task understanding or inattentiveness. This filtered dataset contains 1812 preferences from 50 subjects.

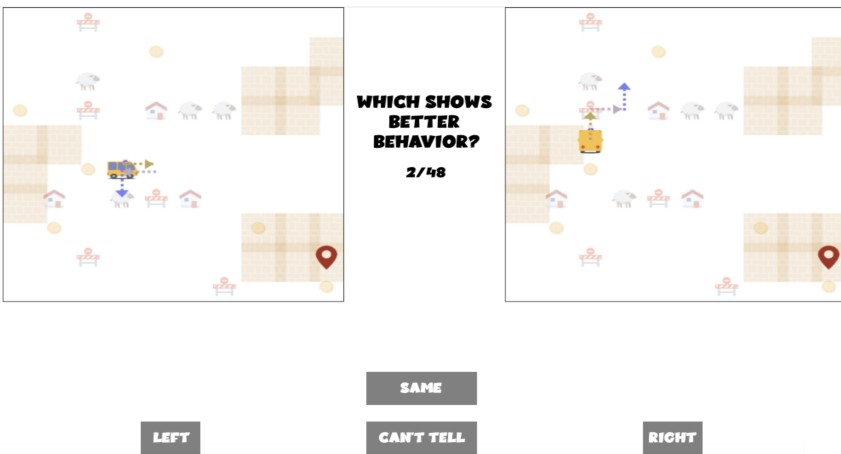

Figure 7: Interface shown to subjects during preference elicitation. The left trajectory shows the yellow van doubling back on itself before hitting a sheep. The right trajectory shows the van hitting a road block.

## 4.3 Selection of segment pairs for labeling

We collected human preferences in two stages, each with different methods for selecting which segment pairs to present for labeling. The sole purpose of collecting this second-stage data was to improve the reward-learning performance of the partial return model, $P_{\Sigma r}$. Without second-stage data, $P_{\Sigma r}$ compared even worse to $P_{regret}$ than in the results described in Section 6, performing worse than a uniformly random policy (see Appendix F.3.3).

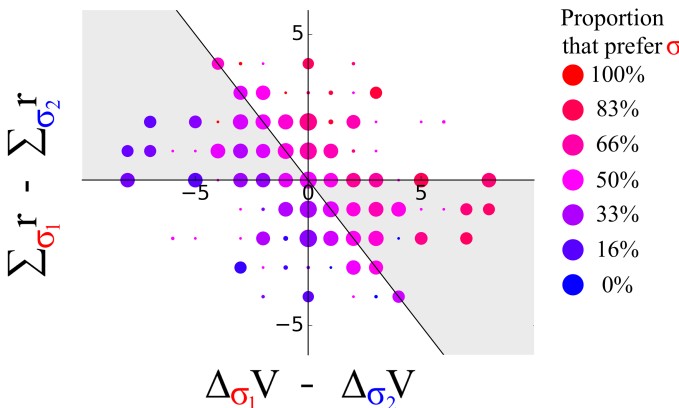

Figure 8: Proportions at which subjects preferred each segment in a pair, plotted by the difference in the segments' changes in state values (x-axis) and partial returns (y-axis). The diagonal line shows points of preference indifference for $P_{regret}$. Points of indifference for $P_\Sigma$ lie on the x-axis. The shaded gray area indicates where the partial return and regret models disagree, each giving a different segment a preference probability greater than 0.5. Each circle's area is proportional to the number of samples it describes. A visual test of which preference model better fits the data is as follows: if the human subjects followed the partial return preference model, the color gradient would be orthogonal to the x-axis. If they followed the regret preference model, the color gradient would be orthogonal to the diagonal line, since regret on this plot is $x+y$. Visual inspection of the color gradient reveals the latter color gradient, suggesting they more closely followed the regret preference model.

Both stages' data are combined and used as a single dataset. These methods and their justification are described in Appendix D.3.

## 5 Descriptive results

This section considers how well different preference models explain our dataset of human preferences.

### 5.1 Correlations between preferences and segment statistics

Recall that with deterministic transitions, the regret of a segment has 3 components: $regret_d(\sigma|\tilde{r}) = V_{\tilde{r}}^*(s_0^\sigma) - (\Sigma_\sigma \tilde{r} + V_{\tilde{r}}^*(s_{|\sigma|}^\sigma))$, one of which is partial return, $\Sigma_\sigma \tilde{r}$. We hypothesize that the other two terms—the values of segments' start and end states, which are included in $P_{regret}$ but not in $P_\Sigma$—affect human preferences, independent of partial return. If this hypothesis is true, then we have more confidence that preference models that include start and end state values will be more effective during inference of the reward functions.

The dataset of preferences is visualized in Figure 8. To simplify analysis, we combine the two parts of $regret_d(\sigma|r)$ that are additional to $\Sigma_\sigma \tilde{r}$ and introduce the following shorthand: $\Delta_\sigma V_{\tilde{r}} \triangleq V_{\tilde{r}}^*(s_{|\sigma|}^\sigma) - V_{\tilde{r}}^*(s_0^\sigma)$.

Note that with an algebraic manipulation (see Appendix E.1), $regret_d(\sigma_2|\tilde{r}) - regret_d(\sigma_1|\tilde{r}) = (\Delta_{\sigma_1} V_{\tilde{r}} - \Delta_{\sigma_2} V_{\tilde{r}}) + (\Sigma_{\sigma_1} \tilde{r} - \Sigma_{\sigma_2} \tilde{r})$. Therefore, on the diagonal line in Figure 8, $regret_d(\sigma_2|r) = regret_d(\sigma_1|r)$, making the $P_{regret_d}$ preference model indifferent. This plot shows how $\Delta_\sigma V_r$ has influence independent of partial return by focusing only on points at a chosen $y$-axis value; if the colors along the corresponding horizontal line reddens as the $x$-axis value increases, then $\Delta_\sigma V_r$ appears to have independent influence.

To statistically test for independent influence of $\Delta_\sigma V_r$ on preferences, we consider subsets of data where $\Sigma_{\sigma_1} r - \Sigma_{\sigma_2} r$ is constant. For $\Sigma_{\sigma_1} r - \Sigma_{\sigma_2} r = -1$ and $\Sigma_{\sigma_1} r - \Sigma_{\sigma_2} r = -2$, the only values with more than 30 samples that also include informative samples with both negative and positive values of $regret(\sigma_1|r) - regret(\sigma_2|r)$, the Spearman's rank correlations between $\Delta_\sigma V_r$ and the preferences are significant ($r >= 0.3$, $p < 0.0001$). This result indicates that $\Delta_\sigma V_r$ *influences human preferences independent of partial return*, validating our hypothesis that humans form preferences based on information about segments' start states and end states, not only partial returns.

## 5.2 Likelihood of human preferences under different preference models

To examine how well each preference model predicts human preferences, we calculate the cross-entropy loss for each model (Equation 1)—i.e., the negative log likelihood—of the preferences in our dataset. Scaling reward by a constant factor does not affect the set of optimal policies. Therefore, throughout this work we ensure that our analyses of preference models are insensitive to reward scaling. To do so for this specific analysis, we conduct 10-fold cross validation to learn a reward scaling factor for each of $P_{regret}$ and $P_{\Sigma r}$. Table 1 shows that the loss of $P_{regret}$ is lower than that of $P_{\Sigma r}$, indicating that $P_{regret}$ is more reflective of how people actually express preferences.

| Preference model | Loss |
|---|---|
| $P(\cdot)=0.5$ (uninformed) | 0.69 |
| $P_{\Sigma r}$ (partial return) | 0.62 |
| $P_{regret}$ | **0.57** |

Table 1: Mean cross-entropy test loss over 10-fold cross validation (n=1812) from predicting human preferences. Lower is better.

## 6 Results from learning reward functions

Analysis of a preference model's predictions of human preferences is informative, but such predictions are a means to the ends of learning human-aligned reward functions and policies. We now examine each preference model's performance in these terms. In all cases, we learn a reward function $\hat{r}$ according to Equation 1 and apply value iteration (Sutton & Barto, 2018) to find the approximately optimal $Q_{\hat{r}}^*$ function. For this $Q_{\hat{r}}^*$, we then evaluate the mean return of the maximum-entropy optimal policy— which chooses uniformly randomly among all *opti-*

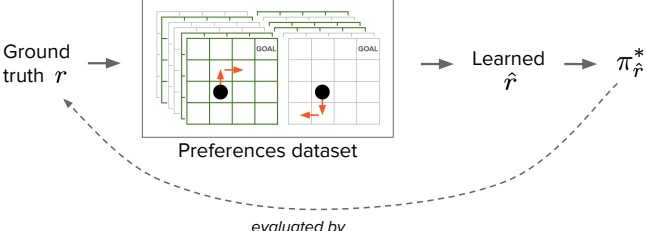

Figure 9: The general design pattern used for learning a reward function from preferences and evaluating that reward function. The generic gridworld shown is for illustrative purposes only.

*mal* actions—with respect to the ground-truth reward function $r$, over $D_0$. This methodology is illustrated in Figure 9.

To compare performance across different MDPs, the mean return of a policy $\pi$, $V_r^\pi$, is normalized to $(V_r^\pi - V_r^U)/V_r^*$, where $V_r^*$ is the optimal expected return and $V_r^U$ is the expected return of the uniformly random policy (both given $D_0$). Normalized mean return above 0 is better than $V_r^U$. Optimal policies have a normalized mean return of 1, and we consider above 0.9 to be *near optimal*.

### 6.1 An algorithm to learn reward functions with $regret(\sigma|\hat{r})$

Algorithm 1 is a general algorithm for learning a *linear* reward function according to $P_{regret}$. This regret-specific algorithm only changes the regret-based algorithm from Section 2.2 by replacing Equation 5 with a tractable approximation of regret, avoiding expensive repeated evaluation of $V_{\hat{r}}^*(\cdot)$ and $Q_{\hat{r}}^*(\cdot,\cdot)$ to compute $P_{regret}(\cdot|\hat{r})$ during reward learning. Specifically, successor features for a set of policies are used to approximate the optimal state values and state-action values for *any* reward function. This algorithm straightforwardly applies general policy iteration (GPI) with successor features to approximate optimal state and action values for arbitrary reward functions, as described by Barreto et al. (2016).

**Approximating $P_{regret}$ with successor features**    Following the notation of Barreto et al., assume the ground-truth reward is linear with respect to a feature vector extracted by $\phi: S \times A \times S \to \mathbb{R}^d$ and a weight vector $\boldsymbol{w_r} \in \mathbb{R}^d$: $r(s,a,s')=\phi(s,a,s')^\top \boldsymbol{w_r}$. During learning, $\boldsymbol{w_{\hat{r}}}$ similarly expresses $\hat{r}$ as $\hat{r}(s,a,s')=\phi(s,a,s')^\top \boldsymbol{w_{\hat{r}}}$.

Given a policy $\pi$, the successor features for $(s,a)$ are the expectation of discounted reward features from that state-action pair when following $\pi$: $\boldsymbol{\psi_Q^\pi}(s,a) = E^\pi[\sum_{t=0}^\infty \gamma^t \phi(s_t, a_t, s_{t+1})|s_0 = s, a_0 = a]$. Therefore, $Q_{\hat{r}}^\pi(s,a)=\boldsymbol{\psi_Q^\pi}(s,a)^\top \boldsymbol{w_{\hat{r}}}$. Additionally, state-based successor features can be calculated from the $\boldsymbol{\psi_Q^\pi}$ above as $\boldsymbol{\psi_V^\pi}(s)=\sum_{a\in A}\pi(a|s)\boldsymbol{\psi_Q^\pi}(s,a)$, making $V_{\hat{r}}^\pi(s)=\boldsymbol{\psi_V^\pi}(s)^\top \boldsymbol{w_{\hat{r}}}$.

Given a set $\Psi_Q$ of state-action successor feature functions and a set $\Psi_V$ of state successor feature functions for various policies and given a reward function via $\boldsymbol{w_{\hat{r}}}$, $Q_{\hat{r}}^{\pi^*}(s,a) \geq max_{\boldsymbol{\psi_Q} \in \Psi_Q}[\boldsymbol{\psi_Q^\pi}(s,a)^\top \boldsymbol{w_{\hat{r}}}]$ and

---

**Algorithm 1** Linear reward learning with regret preference model ($P_{regret}$), using successor features

---

1: Input: a set of policies
2: $\Psi \leftarrow \varnothing$
3: **for** *each reward function policy $\pi_{SF}$ in the input set* **do**
4:     estimate $\psi_Q^{\pi_{SF}}$ and $\psi_V^{\pi_{SF}}$ (if not estimated already during step 4)
5:     add $\psi_Q^{\pi_{SF}}$ to $\Psi_Q$
6:     add $\psi_V^{\pi_{SF}}$ to $\Psi_V$
7: **end for**
8: **repeat**
9:     optimize $\boldsymbol{w}_{\hat{r}}$ by loss of Eqn. 1, calculating $\tilde{P}_{regret}(\sigma_1 \succ \sigma_2 | \hat{r})$ via Eqn. 6, using $\Psi_Q$ and $\Psi_V$
10: **until** *stopping criteria are met*
11: **return** $\boldsymbol{w}_{\hat{r}}$

---

$V_{\hat{r}}^{\pi^*}(s) \geq max_{\boldsymbol{\psi_V} \in \Psi_V}[\boldsymbol{\psi}_V^{\pi}(s)^\top \boldsymbol{w}_{\hat{r}}]$ (Barreto et al., 2016), so we use these two maximizations as approximations of $Q_{\hat{r}}^*(s,a)$ and $V_{\hat{r}}^*(s)$, respectively. In practice, to enable gradient-based optimization with current tools, the maximization in this expression is replaced with the softmax-weighted average, making the loss function linear. Focusing first on the approximation of $V_{\hat{r}}^*(s)$, for each $\boldsymbol{\psi_V} \in \Psi_V$, a softmax weight is calculated for $\boldsymbol{\psi}_V^{\pi}(s)$: $softmax_{\Psi_V}(\boldsymbol{\psi}_V^{\pi}(s)^\top \boldsymbol{w}_{\hat{r}}) \triangleq [(\boldsymbol{\psi}_V^{\pi}(s)^\top \boldsymbol{w}_{\hat{r}})^{1/T}]/[(\sum_{\boldsymbol{\psi'_V} \in \Psi_V} \boldsymbol{\psi'}_V^{\pi}(s)^\top \boldsymbol{w}_{\hat{r}})^{1/T}]$, where temperature $T$ is a constant hyperparameter. The resulting approximation of $V_{\hat{r}}^*(s)$ is therefore defined as $\tilde{V}_{\hat{r}}^*(s) \triangleq \sum_{\boldsymbol{\psi_V} \in \Psi_V} softmax_{\Psi_V}(\boldsymbol{\psi}_V^{\pi}(s)^\top \boldsymbol{w}_{\hat{r}})[\boldsymbol{\psi}_V^{\pi}(s)^\top \boldsymbol{w}_{\hat{r}}]$. Similarly, to approximate $Q_{\hat{r}}^*(s,a)$, $softmax_{\Psi_Q}(\boldsymbol{\psi}_Q^{\pi}(s,a)^\top \boldsymbol{w}_{\hat{r}}) \triangleq [(\boldsymbol{\psi}_Q^{\pi}(s,a)^\top \boldsymbol{w}_{\hat{r}})^{1/T}]/[(\sum_{\boldsymbol{\psi'_Q} \in \Psi} \boldsymbol{\psi'}_Q^{\pi}(s,a)^\top \boldsymbol{w}_{\hat{r}})^{1/T}]$ and $\tilde{Q}_{\hat{r}}^*(s,a) \triangleq \sum_{\boldsymbol{\psi_Q} \in \Psi_Q} softmax_{\Psi_Q}(\boldsymbol{\psi}_Q^{\pi}(s,a)^\top \boldsymbol{w}_{\hat{r}})[\boldsymbol{\psi}_Q^{\pi}(s,a)^\top \boldsymbol{w}_{\hat{r}}]$. Consequently, from Eqns. 4 and 5, the corresponding approximation $\tilde{P}_{regret}$ of the regret preference model is:

$$\tilde{P}_{regret}(\sigma_1 \succ \sigma_2 | \hat{r}) = logistic\left(\sum_{t=0}^{|\sigma_2|-1}\left[\tilde{V}_{\hat{r}}^*(s_t^{\sigma_2}) - \tilde{Q}_{\hat{r}}^*(s_t^{\sigma_2}, a_t^{\sigma_2})\right] - \sum_{t=0}^{|\sigma_1|-1}\left[\tilde{V}_{\hat{r}}^*(s_t^{\sigma_1}) - \tilde{Q}_{\hat{r}}^*(s_t^{\sigma_1}, a_t^{\sigma_1})\right]\right) \quad (6)$$

**The algorithm** In Algorithm 1, lines 8–11 describe the supervised-learning optimization using the approximation $\tilde{P}_{regret}$, and the prior lines create $\Psi_Q$ and $\Psi_V$. Specifically, given a set of input policies (line 1), for each such policy $\pi_{SF}$, successor feature functions $\Psi_Q^{\pi_{SF}}$ and $\Psi_V^{\pi_{SF}}$ are estimated (line 4), which by default would be performed by a minor extension of a standard policy evaluation algorithm as detailed by Barreto et al. (2016). Note that the reward function that is ultimately learned is not restricted to be in the input set of reward functions, which is used only to create an approximation of regret.

One potential source of the input set of policies is a set of reward functions, where each input policy is the result of policy improvement on one reward function. We follow this method in our experiments, randomly generating reward functions and then estimating an optimal policy for each reward function. Specifically, for each reward function, we seek the the maximum entropy optimal policy, which resolves ties among optimal actions in a state via a uniform distribution over those optimal actions.

Further details of our instantiation of Algorithm 1 for the delivery domain can be found in Appendix F.1, along with preliminary guidance for choosing an input set of policies (Appendix F.1.1) and for extending it to reward functions that might be non-linear (Appendix F.1.2).

## 6.2 Results from synthetic preferences

Before considering human preferences, we first ask how each preference model performs when it correctly describes how the preferences in its training set were generated. In other words, we investigate empirically how well the preference model could perform if humans perfectly adhered to it. Recall that the ground-truth reward function, $r$, is used to create these preferences but is inaccessible to the reward-learning algorithms.

For these evaluations, either a stochastic or noiseless preference model acts a preference generator to create a preference dataset. Then the stochastic version of the same model is used for reward learning, which prevents

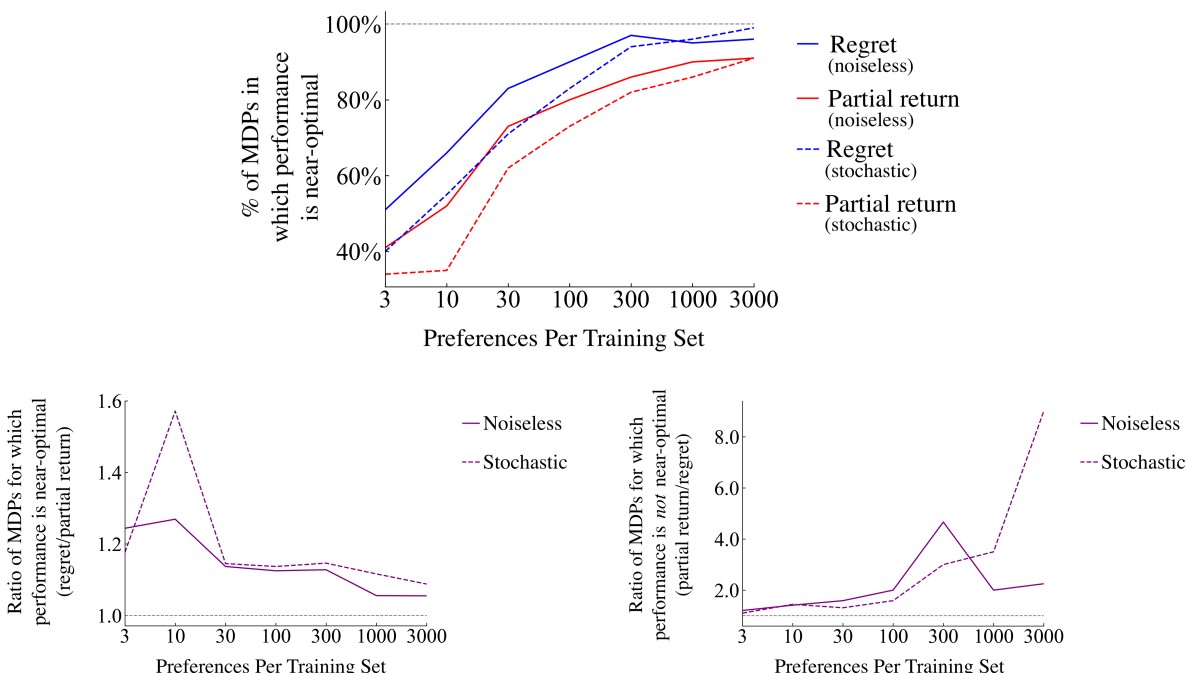

Figure 10: Performance comparison over 100 randomly generated deterministic MDPs when each preference model creates its own training dataset and learns from it. Performance with the regret preference model is consistently better, regardless of training set size or whether preferences are generated stochastically. The bottom-left and bottom-right plots are created from the top plot. The bottom-left plot shows the ratio of between each preference model's success rate. The bottom-right plot shows the ratio between each preference model's rate of failure to reach near-optimal performance. For easier visual comparison, the ratios of each plot are chosen such that higher values indicate better performance by the regret preference model.

the introduction of a hyperparameter. Note that the stochastic preference model can approach determinism through scaling the reward function, so learning a reward function with the stochastic preference model from deterministically generated preferences does not remove our ability to fit a reward function to those preferences. For the noiseless case, the deterministic preference generator compares a segment pair's $\Sigma_\sigma r$ values for $P_{\Sigma r}$ or their $regret(\sigma|r)$ values for $P_{regret}$. Note that through reward scaling the preference generators approach determinism in the limit, so this noiseless analysis examines minimal-entropy versions of the two preference-generating models. (The opposite extreme, uniformly random preferences, would remove all information from preferences and therefore is not examined.) In the stochastic case, for each preference model, each segment pair is labeled by sampling from that preference generator's output distribution (Eqs 2 or 5), using the unscaled ground-truth reward function.

We created 100 deterministic MDPs that instantiate variants of our delivery domain (see Section 4.1). To create each MDP, we sampled from sets of possible widths, heights, and reward component values, and the resultant grid cells were randomly populated with a destination, objects, and road surface types (see Appendix F.2 for details). Each segment in the preference datasets for each MDP was generated by choosing a start state and three actions, all uniformly randomly. For a set number of preferences, each method had the same set of segment pairs in its preference dataset. Figure 10 shows the percentage of MDPs in which each preference model results in near-optimal performance. The regret preference model outperforms the partial return model at every dataset size, both with and without noise. By a Wilcoxon paired signed-rank test on normalized mean returns, $p < 0.05$ for 86% of these comparisons and $p < 0.01$ for 57% of them, as reported in Appendix F.2.

Further analyses can be found in Appendix F.2: with stochastic transitions, with different segment lengths, without segments that terminate before their final transition, and with additional novel preference models.

### 6.3 Results from human preferences

We now consider the reward-learning performance of each preference model on preferences generated by humans for our specific delivery task. We randomly assign human preferences from our gathered dataset to different numbers of same-sized partitions, resulting in different training set sizes, and test each preference model on each partition. Figure 11 shows the results. With smaller training sets (20–100 partitions), the regret preference model results in near-optimal performance more often. With larger training sets (1–10 partitions), both preference models always reach near-optimal return, but the mean return from the regret preference model is higher for all of these partitions except for only 3 partitions in the 10-partition test. Applying a Wilcoxon paired signed-rank test on normalized mean return to each group with 5 or more partitions, $p < 0.05$ for all numbers of partitions except 100 and $p < 0.01$ for 20 and 50 partitions. To summarize, we find that both the regret and the partial return preference models achieve near-optimal performance when the dataset is sufficiently large—although the performance of the regret preference model is nonetheless almost always higher—and we also find that regret achieves near-optimal performance more often with smaller datasets.

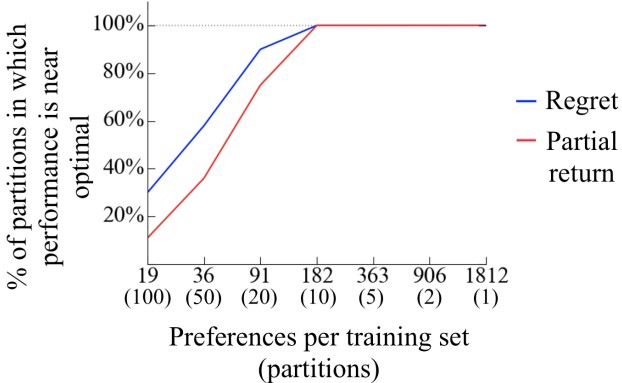

Figure 11: Performance comparison over various amounts of human preferences. Each partition has the number of preferences shown or one less.

Using the human preferences dataset, Appendix F.3 contains further analyses: learning without segments that terminate before their final transition, learning via additional novel preference models, and testing the learned reward functions on other MDPs with the same ground-truth reward function.

## 7 Conclusion

Over numerous evaluations with human preferences, our proposed regret preference model ($P_{regret}$) shows improvements summarized below over the previous partial return preference model ($P_{\Sigma r}$). When each preference model generates the preferences for its own infinite and exhaustive training set, we prove that $P_{regret}$ identifies the set of optimal policies, whereas $P_{\Sigma r}$ is not guaranteed to do so in multiple common contexts. With finite training data of synthetic preferences, $P_{regret}$ also empirically results in learned policies that tend to outperform those resulting from $P_{\Sigma r}$. This superior performance of $P_{regret}$ is also seen with human preferences. In summary, our analyses suggest that regret preference models are more effective both descriptively with respect to human preferences and also normatively, as the model we want humans to follow if we had the choice.

Independent of $P_{regret}$, this paper also reveals that segments' changes in state values provide information about human preferences that is not fully provided by partial return. More generally, we show that the choice of preference model impacts the performance of learned reward functions.

This study motivates several new directions for research. Future work could address any of the limitations detailed in Appendix A.1. Specifically, future work could further test the general superiority of $P_{regret}$ or apply it to deep learning settings. Additionally, *prescriptive* methods could be developed via the subject interface or elsewhere to nudge humans to conform more to $P_{regret}$ or to other normatively appealing preference models. Lastly, this work provided conclusive evidence that the choice of preference model is impactful. Subsequent efforts could seek preference models that are even more effective with preferences from actual humans.

### Acknowledgements

We thank Jordan Schneider, Garrett Warnell, Ishan Durugkar, and Sigurdur Orn Adalgeirsson, who each gave extensive and insightful feedback on earlier drafts. This work has taken place in part in the Safe, Correct, and Aligned Learning and Robotics Lab (SCALAR) at The University of Massachusetts Amherst. SCALAR research is supported in part by the NSF (IIS-1749204, IIS-1925082), AFOSR (FA9550-20-1-0077), and ARO

(78372-CS, W911NF-19-2-0333). This work has also taken place in part in the Learning Agents Research Group (LARG) at UT Austin. LARG research is supported in part by NSF (FAIN-2019844), ONR (N00014-18-2243), ARO (W911NF-19-2-0333), DARPA, Bosch, and UT Austin's Good Systems grand challenge. Peter Stone serves as the Executive Director of Sony AI America and receives financial compensation for this work. The terms of this arrangement have been reviewed and approved by the University of Texas at Austin in accordance with its policy on objectivity in research.

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

After Appendix A, the appendix is organized according to the major sections and subsections of the main content.

# A    Limitations and ethics

## A.1    Limitations

Some limitations of the regret preference model are discussed in the paragraph "Regret as a model for human preference" in Section 2.2, including assumptions that a person giving preferences can distinguish between optimal and suboptimal segments, that they follow a Boltzmann distribution (i.e., a Luce Shepard choice rule), and that they base their preferences on decision quality even when transition stochasticity results in segment pairs for which the worse decision has a better outcome.

Our proposed algorithm (Section 6.1) has a few additional limitations. Generating candidate successor features for the approximations $\tilde{Q}_{\hat{r}}^*$ and $\tilde{V}_{\hat{r}}^*$ may be difficult in complex domains. Specifically, challenges include choosing the set of policies or reward functions for which to compute successor features (line 3 of Algorithm 1, discussed in Appendix F.1.1) and creating a reward feature vector $\phi$ for non-linear reward functions (discussed in Appendix F.1.2). Additionally, although learning with $P_{regret}$ is more sample efficient in our experiments, it is computationally slower than learning with $P_{\Sigma r}$ because of the additional need to compute successor features and the use of the softmax function to approximate $Q_{\hat{r}}^*$ and $V_{\hat{r}}^*$. Nonetheless, we may accept the tradeoff of an increase in computational time that reduces the number of human samples needed or that improves the reward function's alignment with human stakeholders' interests. Lastly, the loss during optimization with $P_{regret}$ was unstable, which we addressed by taking the minimum loss over all epochs during training. Therefore, for more complex reward feature vectors ($\phi$) than our 6-element vector for the delivery task, extra care might be needed to avoid overfitting $\hat{r}$, for example by withholding some preference data to serve as a test set.

We also generally assume that the RL algorithm and reward learning algorithm use the same discount factor as in the MDP$\backslash r$ specification. One weakness of contemporary deep RL is that RL algorithms require artificially lower discount factors than the true discount factor of the task. The interaction of this discounting with preference models is considered in Appendix F.2. Our expectation though is that this weakness of deep RL algorithms is likely a temporary one, and so we focused our analysis on simple tasks in which we do not need to artificially lower the RL algorithm's discount factor. However, further investigation of the interaction between preference models and discount factors would aid near-term application of $P_{regret}$ to deep RL domains.

This work also does not consider which segment pairs should be presented for labeling with preferences used for reward learning. However, other research has addressed this problem through active learning (Lee et al., 2021a; Christiano et al., 2017; Akrour et al., 2011), and future work could consider the compatibility of these active learning methods with our Algorithm 1, combining the improved sample efficiency of $P_{regret}$ with that of these active learning methods.

Additionally, all preference models considered in this paper are evaluated with the assumption that for any pair of segments in the preference dataset, both segments have the same length. However, for segment pairs of different lengths, the regret preference model may act in ways that violate our intuition for how humans would give preferences. In particular, a short, highly suboptimal segment might be more preferable under the regret preference model than a much longer segment that is near-optimal yet nonetheless has a higher regret, since the regret of a segment is the sum of the regret of its individual transitions. The partial return model—which already breaks our intuition with same-length segment pairs—suffers from a similar limitation, in that it further deviates from intuition as the segment lengths diverge. Allowing such different-length segment pairs appears uncommon in practice but may be difficult to avoid in some task domains.

Regarding the human side of the problem of reward learning from preferences, further research could provide several improvements. First, we are confident that humans can be influenced by their training and by the preference elicitation interface, which is a particularly rich direction for follow-up study. We also do not consider how to handle learning reward functions from multiple human stakeholders who have different preferences, a topic we revisit in Appendix A.2. Lastly, we expect humans to deviate from any simple model, including $P_{regret}$,

and a fine-grained characterization of how humans generate preferences could produce preference models that further improve the alignment of the reward functions that are ultimately learned from human preferences.

## A.2 Ethical statement

This work is meant to address ethical issues that arise when autonomous systems are deployed without properly aligning their objectives with those of human stakeholders. It is merely a step in that direction, and overly trusting in our methods—even though they improve on previous methods for alignment—could result in harm caused by poorly aligned autonomous systems.

When considering the objectives for such systems, a critical ethical question is *which* human stakeholders' interests the objectives should be aligned with and how multiple stakeholders' interests should be combined into a single objective for an autonomous system. We do not address these important questions, instead making the convenient-but-flawed assumption that many different humans' preferences can simply be combined. In particular, care should be taken that vulnerable and marginalized communities are adequately represented in any technique or deployment to learn a reward function from human preferences in high-impact settings. The stakes are high: for example, a reward function that is only aligned with a corporation's financial interests could lead to exploitation of such communities or more broadly to exploitation of or harm to users.

In this specific work, our filter for which Mechanical Turk Workers could join our study is described in Appendix D. We did not gather demographic information and therefore we cannot assess how representative our subjects are of any specific population.

## A.3 On the challenge of using regret preference models in practice

We have provided evidence—theoretically and with experimentation—that the regret preference model is more effective when precisely measured or effectively approximated. The challenge of efficiently creating such approximations presents one clear path for future research. We believe this challenge does not justify staying within the local maximum of the partial return preference model.

Like the regret preference model, inverse reinforcement learning (IRL) was founded on an algorithm that requires solving an MDP in an inner loop of learning a reward function. For example, see the seminal work on IRL by Ng & Russell (2000). IRL has been an impactful problem despite this challenge, and handling this inner-loop computational demand is the focus of much IRL research.

Future work on the application of the regret preference model can face the challenge of scaling to more complex problems. Given that IRL has made tremendous progress in this direction and Brown et al. (2020) have scaled an algorithm with similar needs to those of Algorithm 1, we are optimistic that the methods to scale can be developed, likely with light adaptation from existing methods (e.g., in Brown et al. or in Appendix F.1.1 and F.1.2).

# B Preference models for learning reward functions

Here we extend the content of Section 2, focusing on preference models and learning algorithms that use them. This corresponding section of the appendix provides a simple derivation of the logistic form of these preference models, discusses extensions of the regret preference model, sketches an alternative way to learn a policy with it, and discusses the relationship of inverse reinforcement learning to learning reward functions with a regret preference model.

## B.1 Derivation of the logistic expression of the Boltzmann distribution

For the reader's convenience, below we derive the logistic expression of a function that is based on two subtracted values from the Boltzmann distribution (i.e., softmax) representation that is more common in past work. These values are specifically the same function $f$ applied to each segment, which is a general expression of both of the

preference models considered here.

$$P(\sigma_1 \succ \sigma_2) = \frac{\exp\left[f(\sigma_1)\right]}{\exp\left[f(\sigma_1)\right] + \exp\left[f(\sigma_2)\right]}$$

$$= \frac{1}{1 + \frac{\exp\left[f(\sigma_2)\right]}{\exp\left[f(\sigma_1)\right]}}$$

$$= \frac{1}{1 + \exp\left[f(\sigma_2) - f(\sigma_1)\right]}$$

$$= logistic(f(\sigma_1) - f(\sigma_2)). \tag{7}$$

## B.2 Learning reward functions from preferences, with discounting

For the equations from the paper's body that assume that there is no temporal discounting (i.e., $\gamma = 1$), we share in this section versions that do not make this assumption. If $\gamma = 1$, then the equations below simplify to those in the body of the paper. To allow for fully myopic discounting with $\gamma = 0$, we define $0^0 = 1$.

Recall that $\tilde{r}$ indicates an arbitrary reward function, which may not be the ground-truth reward function, $r$, and $\hat{r}$ refers to a learned reward function. Similarly, $\tilde{\gamma}$ refers to an arbitrary exponential discount factor, which may not be the ground-truth discount factor, $\gamma$, and $\hat{\gamma}$ refers to the discount factor during learning, which could be inferred or hand-coded. Also, the notation of $V_{\tilde{r}}^*$ and $Q_{\tilde{r}}^*$ are expanded in this subsection to denote the discounting in their expected return: $V_{(\tilde{r},\tilde{\gamma})}^*$ and $Q_{(\tilde{r},\tilde{\gamma})}^*$, respectively.

In most of this article, the discount factor used during reward function inference is hard-coded as $\hat{\gamma} = 1$. However, in the theory of Section 3, we assume $\gamma$ is *not* known to reach more general conclusions. In this subsection, for generality we likewise assume that $\gamma$ is not known, using $\tilde{\gamma}$ generally and using $\hat{\gamma}$ in notation we consider specific to reward function inference.

The discounted versions of the preference models below rely on a cross entropy loss function that is identical to Equation 1 except for the inclusion of discounting notation:

$$loss(\hat{r}, \hat{\gamma}, D_\succ) = -\sum_{(\sigma_1, \sigma_2, \mu) \in D_\succ} \mu_1 \log P(\sigma_1 \succ \sigma_2 | \hat{r}, \hat{\gamma}) + \mu_2 \log P(\sigma_1 \prec \sigma_2 | \hat{r}, \hat{\gamma}) \tag{8}$$

**Partial return**   With discounting, the **partial return** of a segment $\sigma$ is $\sum_{t=0}^{|\sigma|-1} \gamma^t \tilde{r}_{\sigma,t}$. This notation differs from that in Section 2.1 in that the subscript of the reward symbol $\tilde{r}_{\sigma,t}$ is now expanded to include which segment it comes from.

The preference model based on partial return *with exponential discounting* is expressed below, generalizing Equation 2.

$$P_{\Sigma r}(\sigma_1 \succ \sigma_2 | \tilde{r}, \tilde{\gamma}) = logistic\left(\sum_{t=0}^{|\sigma_1|-1} \tilde{\gamma}^t \tilde{r}_{\sigma_1,t} - \sum_{t=0}^{|\sigma_2|-1} \tilde{\gamma}^t \tilde{r}_{\sigma_2,t}\right). \tag{9}$$

**Regret**   With discounting, for a transition $(s_t, a_t, s_{t+1})$ in a segment containing only deterministic transitions, $regret_{\mathrm{d}}(\sigma_t | \tilde{r}, \tilde{\gamma}) \triangleq V_{(\tilde{r},\tilde{\gamma})}^*(s_t^\sigma) - [\tilde{r}_t + \tilde{\gamma} V_{(\tilde{r},\tilde{\gamma})}^*(s_{t+1}^\sigma)]$.

For a full deterministic segment, $regret_d(\cdot | \tilde{r}, \tilde{\gamma})$ with exponential discounting is defined as follows, generalizing Equation 3.

$$regret_d(\sigma | \tilde{r}, \tilde{\gamma}) \triangleq \sum_{t=0}^{|\sigma|-1} \tilde{\gamma}^t regret_d(\sigma_t | \tilde{r}, \tilde{\gamma})$$

$$= V_{(\tilde{r},\tilde{\gamma})}^*(s_0^\sigma) - \left(\sum_{t=0}^{|\sigma|-1} \tilde{\gamma}^t \tilde{r}_{\sigma,t} + \tilde{\gamma}^{|\sigma|} V_{(\tilde{r},\tilde{\gamma})}^*(s_{|\sigma|}^\sigma)\right), \tag{10}$$

Like Equation 3, this discounted form of deterministic regret also measures how much the segment reduces expected return from the start state value, $V_{(\tilde{r},\tilde{\gamma})}^*(s_0^\sigma)$.

To create the general expression of discounted regret that accounts for potential stochastic transitions, we note that, with discounting, the effect on expected return of transition stochasticity from a transition $(s_t, a_t, s_{t+1})$ is $[\tilde{r}_t + \tilde{\gamma} V^*_{(\tilde{r}, \tilde{\gamma})}(s_{t+1})] - Q^*_{(\tilde{r}, \tilde{\gamma})}(s_t, a_t)$ and add this expression once per transition to get $regret(\sigma | \tilde{r}, \tilde{\gamma})$, removing the subscript $d$ that refers to determinism. The discounting does not change the simplified expressions in Equation 4, the regret for a single transition:

$$
\begin{aligned}
regret(\sigma_t | \tilde{r}, \tilde{\gamma}) &= [V^*_{(\tilde{r}, \tilde{\gamma})}(s_t^\sigma) - [\tilde{r}_t + \tilde{\gamma} V^*_{(\tilde{r}, \tilde{\gamma})}(s_{t+1}^\sigma)]] + [[\tilde{r}_t + \tilde{\gamma} V^*_{(\tilde{r}, \tilde{\gamma})}(s_{t+1}^\sigma)] - Q^*_{(\tilde{r}, \tilde{\gamma})}(s_t^\sigma, a_t^\sigma)] \\
&= V^*_{(\tilde{r}, \tilde{\gamma})}(s_t^\sigma) - Q^*_{(\tilde{r}, \tilde{\gamma})}(s_t^\sigma, a_t^\sigma) \\
&= -A^*_{(\tilde{r}, \tilde{\gamma})}(s_t^\sigma, a_t^\sigma).
\end{aligned}
\tag{11}
$$

With both discounting and accounting for potential stochastic transitions, regret for a full segment is

$$
\begin{aligned}
regret(\sigma | \tilde{r}, \tilde{\gamma}) &= \sum_{t=0}^{|\sigma|-1} \tilde{\gamma}^t regret(\sigma_t | \tilde{r}, \tilde{\gamma}) \\
&= \sum_{t=0}^{|\sigma|-1} \tilde{\gamma}^t \left[ V^*_{(\tilde{r}, \tilde{\gamma})}(s_t^\sigma) - Q^*_{(\tilde{r}, \tilde{\gamma})}(s_t^\sigma, a_t^\sigma) \right] \\
&= \sum_{t=0}^{|\sigma|-1} -\tilde{\gamma}^t A^*_{(\tilde{r}, \tilde{\gamma})}(s_t^\sigma, a_t^\sigma).
\end{aligned}
\tag{12}
$$

The expression of regret above is the most general in this paper and can be used in Equation 5 identically as can the undiscounted version in Equation 4.

Equation 6, the approximation $\tilde{P}_{regret}$ of the regret preference model derived in Section 6.1, is expressed with discounting below.

$$
\tilde{P}_{regret}(\sigma_1 \succ \sigma_2 | \hat{r}, \hat{\gamma}) = logistic\left( \sum_{t=0}^{|\sigma_2|-1} \hat{\gamma}^t \left[ \tilde{V}^*_{(\hat{r}, \hat{\gamma})}(s_t^{\sigma_2}) - \tilde{Q}^*_{(\hat{r}, \hat{\gamma})}(s_t^{\sigma_2}, a_t^{\sigma_2}) \right] - \sum_{t=0}^{|\sigma_1|-1} \hat{\gamma}^t \left[ \tilde{V}^*_{(\hat{r}, \hat{\gamma})}(s_t^{\sigma_1}) - \tilde{Q}^*_{(\hat{r}, \hat{\gamma})}(s_t^{\sigma_1}, a_t^{\sigma_1}) \right] \right)
\tag{13}
$$

Note that the successor features used in Section 6.1 to determine these approximations, $\tilde{V}^*_{(\hat{r}, \hat{\gamma})}$ and $\tilde{Q}^*_{(\hat{r}, \hat{\gamma})}$, already include discounting.

As with the undiscounted versions of the above equations, if two segments have deterministic transitions, end in terminal states, and have the same starting state, this regret model reduces to the partial return model: $P_{regret}(\cdot | \tilde{r}, \tilde{\gamma}) = P_{\Sigma r}(\cdot | \tilde{r}, \tilde{\gamma})$.

**If hard-coding $\hat{\gamma}$, when to set $\hat{\gamma} < 1$ during reward function inference** In reinforcement learning, both $\gamma$ and $r$ together determine the set of optimal policies. Changing either $\gamma$ or $r$ while holding the other constant will often change the set of optimal policies.

For both preference models, we suspect that learning would benefit from using the same discounting during reward inference as the human used while evaluating segments to provide preferences (i.e., setting $\hat{\gamma} = \gamma$. And this same $\hat{\gamma}$ would be used for learning a policy from the learned reward function. On the other hand, when $\hat{\gamma}$ is hand-coded and $\hat{\gamma} \neq \gamma$, the reward inference algorithm will regardless attempt to find an $\hat{r}$ that explains those preferences; however, a set of optimal policies is determined by a reward function *with the discount*, and the set of optimal polices created by the human's reward function and discounting may not be determinable under a different discounting.

Not only is a specific human rater's $\gamma$ unobservable, but psychology and economics researchers have firmly established that humans do not typically follow exponential discounting (Frederick et al., 2002), which should evoke skepticism for hard-coding $\hat{\gamma} < 1$ during reward function inference. One exception is humans who have been trained to apply exponential discounting, such as in certain financial settings. The best model for how humans discount future rewards and punishments is not settled, but one popular model is hyperbolic discounting. Some exploration of RL with hyperbolic discounting exists, including approximating hyperbolically discounted value

function using a mixture of exponentially discounted value functions (Kurth-Nelson & Redish, 2009; Redish & Kurth-Nelson, 2010). However, it has not found clear usage beyond as an auxiliary task to aid representation learning (Fedus et al., 2019). The interpretation of human preferences over segments appears to us to be a strong candidate for using these methods to approximate hyperbolic discounting.

This research topic currently lacks a rigorous treatment of discounting when learning reward functions from human preferences and such an investigation is beyond our scope, and so we leave our guidance above as speculative.

### B.3 Logistic-linear preference model

In Appendices E.2, F.2.5, and F.3.2 we also consider preference models that arise by making the noiseless preference model a linear function over the 3 components of $P_{regret_d}$. Building upon Equation 7 above, we set $f(\sigma) = \vec{w} \cdot \langle V_{\tilde{r}}^*(s_0^\sigma), \Sigma_\sigma, V_{\tilde{r}}^*(s_{|\sigma|}^\sigma) \rangle$. This preference model, $P_{log-lin}$, can be expressed after algebraic manipulation as

$$P_{log-lin}(\sigma_1 \succ \sigma_2 | \tilde{r}) = logistic\left( \vec{w} \cdot \langle V_{\tilde{r}}^*(s_0^{\sigma_1}) - V_{\tilde{r}}^*(s_0^{\sigma_2}), \Sigma_{\sigma_1}\tilde{r} - \Sigma_{\sigma_2}\tilde{r}, V_{\tilde{r}}^*(s_{|\sigma_1|}^{\sigma_1}) - V_{\tilde{r}}^*(s_{|\sigma_2|}^{\sigma_2}) \rangle \right). \quad (14)$$

This logistic-linear preference model is a generalization of $P_{\Sigma r}$ and also of $P_{regret_d}$, the regret preference model for deterministic transitions. Specifically, if $\vec{w} = \langle 0,1,0 \rangle$, then $P_{log-lin}(\cdot|\tilde{r}) = P_{\Sigma r}(\cdot|\tilde{r})$. And if $\vec{w} = \langle -1,1,1 \rangle$, then $P_{log-lin}(\cdot|\tilde{r}) = P_{regret_d}(\cdot|\tilde{r})$. More generally, for some constant $c$, $\vec{w} = \langle 0,c,0 \rangle$ and $\vec{w} = \langle -c,c,c \rangle$ recreate $P_{\Sigma r}$ and $P_{regret_d}$ respectively but with different reward function scaling, which is the same as allowing a different temperature in the Boltzmann distribution that determines preference probabilities. In Appendix E.2, we fit $\vec{w}$ to maximize the likelihood of the human preference dataset under $P_{log-lin}(\cdot|r)$, using the ground-truth $r$, and compare the learned weights to those of $P_{\Sigma r}$ and $P_{regret_d}$.

### B.4 Adding a constant probability of uniformly distributed preference

Appendix E.2 also considers adaptations of $P_{\Sigma r}$, $P_{regret_d}$, and $P_{log-lin}$ that add a constant probability of uniformly distributed preference, as was done by Christiano et al. (2017). The body of the paper does not consider these adaptations.

We create this adaptation, which we will call $P'$ here, from another preference model $P$ by $P'(\sigma_1 \succ \sigma_2) = [(1 - logistic(c)) * P(\sigma_1 \succ \sigma_2)] + [logistic(c)/2]$, where $c$ is a constant that in practice we fit to data and $logistic(c)$ is the constant probability of uniformly random preference. The $logistic(c)$ allows any constant $c$ to result in a the constant probability of uniformly distributed preference to be in $(0,1)$. The term $logistic(c)/2$ gives half of the constant probability to $\sigma_1$ and half to $\sigma_2$. The term $[1 - logistic(c)]$ scales the $P(\sigma_1 \succ \sigma_2)$ probability—which could be $P_{\Sigma r}$, $P_{regret_d}$, or $P_{log-lin}$—to a proportion of the remaining probability. The only difference in this adaptation and Christiano et al.'s 0.1 probability of uniformly distributed preference is that we learn the value of $c$ from training data (in a k-fold cross-validation setting), as we see in Appendix E, whereas Christiano et al. do not share how 0.1 was chosen.

### B.5 Expected return preference model

In Appendix F.3, we test reward learning on a third preference model. This expected return preference model is derived by making $f(\sigma) = -(\Sigma_\sigma \tilde{r} + V_{\tilde{r}}^*(s_{|\sigma|}^\sigma))$, in Equation 7. This segment statistic $f(\sigma)$ can be considered be in between deterministic regret (Equation 3) and partial return, differing from each by one term.

We include this preference model because judging by expected return is intuitively appealing in that it considers the partial return along the segment and the end state value of the segment, and we found it plausible that human preference providers might tend to ignore start state value, as this preference model does. However, reward learning with the regret model outperforms or matches that by this expected return preference model, as we show in Appendix F.3.

### B.6 Relationship to inverse reinforcement learning

Like learning reward functions from pairwise preferences, inverse reinforcement learning (IRL) also involves learning a reward function. However, the inputs to IRL and learning reward functions from pairwise preferences are different: IRL requires demonstrations, not preferences over segment pairs. However, because a a regret-based preference model always prefers optimal segments over suboptimal segments, at least one further connection can be made. If one assumes that a demonstrated trajectory segment is noiselessly optimal—as in the foundational IRL paper on apprenticeship learning (Abbeel & Ng, 2004)—then such a demonstration is equivalent to expressing preference or indifference for the demonstrated segment over all other segments. In other words, no other segment is preferred over the demonstrated segment. However, IRL has its own identifiability issues in noiseless settings (e.g., see Kim et al. (2021)) that, viewed from the lens of preferences, come in part from the "indifference" part of the above statement: since there can be multiple optimal actions from a single state, it is not generally correct to assume that a demonstration of one such action shows a preference over all others, and therefore it remains unclear in IRL what other actions are optimal. Note that since partial-return-based preferences can prefer suboptimal segments over optimal segments, the common assumption in IRL that demonstrations are optimal does not map as cleanly to partial-return-based preferences.

The regret preference model also relates to IRL in that the most basic version of IRL requires solving an MDP in the inner loop (see Algorithm 1 in the survey of IRL by Arora & Doshi (2021)), as appears necessary for a perfect measure of regret while learning a reward function. The progress that IRL has made addressing this challenge gives us optimism that it is similarly addressable for complex tasks in for our proposed algorithm. We discuss potential solutions in Appendix F.1.1 and F.1.2.

## C Theoretical comparisons

**The relevance of noiseless preference generators**    Because we model preferences as stochastic in Section 2, one might reasonably wonder how the above theoretical analysis of noiseless preference generators are relevant. We offer four arguments below.

First, having structured noise provides information that can help both preference models, but these proofs show that there are cases where the signal behind the noise—either regret or partial return—is not sufficient in the partial return case to identify an equivalent reward function. So, in a rough sense, regret more effectively uses both the signal and the noise, which might explain its superior sample efficiency in our experiments across both human labels and synthetic labels. Relatedly, the noiseless setting can help us understand each preference model's sample efficiency in a low-noise setting.

Second, noiseless preferences are also feasible, even if they are rare. Therefore, understanding what can be learned from them is worthwhile. Theorem 3.2 shows that there are MDPs in which there is *no* class of preference models—stochastic or deterministic—that can identify an equivalent reward function from partial-return-based preferences if the preference generator noiselessly prefers according to partial return. Specifically, we show that the mapping from two reward functions with different sets of optimal policies to partial-return based preferences is a many-to-one-mapping, and therefore the information simply does not exist to invert that mapping and identify a reward function with the same set of optimal policies. In contrast, Theorem 3.1 shows that preferences generated noiselessly (and in certain stochastic settings) by regret do not have this issue.

Third, noise is often motivated as modeling human error. Having an algorithm rely on noise—structured in a very specific, Boltzmann-rational way—is an undesirable crutch. Skalse et al. (2022) justify including noiseless preferences in their examinations of identifiability with a similar argument: "these invariances rely heavily on the precise structure of the decision noise revealing cardinal information in the infinite-data limit".

Beyond the work of Skalse et al. (2022), there is broader precedent for considering noiseless human input for theory or derivations. For instance, the foundational IRL research on apprenticeship learning (Abbeel & Ng, 2004) treats demonstrations as noiselessly optimal. Recent work by Kim et al. (2021) focuses on reward identifiability with noiseless, optimal demonstrations.

## D   Additional information for creating a human-labeled preference dataset

### D.1   The preference elicitation interface and study overview

Here we share miscellaneous details about the preference elicitation interface from which we collected human subjects' preferences. This description builds on Section 4.2.

In selecting preferences, subjects had four options. They could prefer either trajectory (left or right), or they could express their preference to be the same or indistinguishable. To provide these preferences, subjects could either click on each of the buttons labeled "LEFT", "RIGHT", "SAME", or "CAN'T TELL" (shown in Figure 7) or by using the arrow keys to select amongst these choices.

For the interface, all icons used to visualize the task were obtained from icons8.com under their Paid Universal Multimedia Licensing Agreement.

We paid all subjects $5 per experiment (i.e., for each a Mechanical Turk HIT), which was chosen using the median time subjects took during a pilot study and then calculating the payment to result in $15 USD per hour. This hourly rate of $15 was chosen because it is commonly recommended as an improved US federal minimum wage. The human subject experiments cost $2,145 USD in total.

An experimental error resulted in the IRB-approved consent form not being presented to human subjects after Mechanical Turk Workers accepted our study. We reported this error to our IRB and received their approval to use the data.

### D.2   Filtering subject data

To join our study via Amazon Mechancial Turk, potential subjects had to meet the following criteria. They had to be located in the United States, have an approval rating of at least 99%, and have completed at least 100 other MTurk HITs. We selected these criteria to improve the probability of collecting data from subjects who would attentively engage with our study and who would understand our training protocol.

We assessed each subject's understanding of the delivery domain and filtered out those who did not comprehend the task, as described below. Specifically, subjects completed a task-comprehension survey, through which we assigned them a task-comprehension score. The questions and answer choices are shown in Table 2. Each fully correct answer was worth 1 point and each partially correct answer was worth 0.5 points. Task-comprehension scores were bounded between 0 and 7. We removed the data from subjects who scored below a threshold of 4.5. The threshold of 4.5 was chosen based on visual analysis of a histogram of scores, attempting to balance high standards for comprehension with retaining sufficient data for analysis.

In addition to filtering based off the task comprehension survey, we also removed a subject's data if they ever preferred colliding the vehicle into a sheep over not doing so. Since such collisions are highly undesirable in this task, we interpreted this preference as evidence of either poor task understanding or inattentiveness.

In total, we collected data from 143 subjects. Data from 58 of these subjects were removed based on subjects' responses to the survey. From what remained, data from another 35 subjects were removed for making the aforementioned sheep-collision preference errors. After this filtering, data from 50 subjects remained. This filtered data consists of 1812 preferences over 1245 unique segment pairs and is used in this article's experiments.

Regarding potential risks to subjects, this data collection had limited or no risk. No offensive content was shown to subjects while they completed the HIT. Mechanical Turk collected Worker IDs, which were used only to link preference data with the results from the task-comprehension survey for filtering data (see Appendix D.2) and then were deleted from our data. No other potentially personally identifiable information was collected.

### D.3   The two stages of data collection

We collected the human preference dataset in two stages, as mentioned in Section 4.2. Here we provide more detail on each stage. These stages differed largely by their goals for data collection and, following those goals, how we chose which segment pairs were presented to subjects for their preference.

Table 2: The task comprehension survey, designed to test participant's comprehension of the domain for the purpose of filtering data. Each full credit answer earned 1 point; each partial credit answer earned 0.5 points. We discarded the data of participants who scored less than 4.5 points overall.

| Question | Full credit answer | Partial credit answer | Other answer choices |
|---|---|---|---|
| What is the goal of this world? (Check all that apply.) | • To maximize profit | • To get to a specific location. 
 • To maximize profit 

 Partial credit was given if both answers were selected. | • To drive as far as possible to explore the world. 
 • To collect as many coins as possible. 
 • To collect as many sheep as possible. 
 • To drive sheep to a specific location. |
| What happens when you run into a house? (Check all that apply.) | • You pay a gas penalty. 
 • You can't run into a house; the world doesn't let you move into it. 

 Full credit was given if both answers were selected. | • You pay a gas penalty. 
 • You can't run into a house; the world doesn't let you move into it. 

 Partial credit was given if only one answer was selected. | • The episode ends. 
 • You get stuck. 
 • To collect as many sheep as possible. |
| What happens when you run into a sheep? (Check all that apply.) | • The episode ends. 
 • You are penalized for running into a sheep. 

 Full credit was given if both answers were selected. | • The episode ends. 
 • You are penalized for running into a sheep. 

 Partial credit was given if only one answer was selected. | • You are rewarded for collecting a sheep. |
| What happens when you run into a roadblock? (Check all that apply.) | • You pay a penalty. | | • The episode ends. 
 • You get stuck. 
 • You can't run into a roadblock; the world doesn't let you move into it. |
| Is running into a roadblock ever a good choice in any town? | • Yes, in certain circumstances. | | • No. |
| What happens when you go into the brick area? (Check all that apply.) | • You pay extra for gas. | | • The episode ends. 
 • You get stuck in the brick area. 
 • You can't go into the brick area; the world doesn't let you move into it. |
| Is entering the brick area ever a good choice? | • Yes, in certain circumstances | | • No |

**First stage**  Figure 12 illustrates the coordinates that segment pairs were sampled from in the first stage of data collection, varying by state value differences and by differences in partial returns over the segments. We sought a range of points that would allow a characterization of human preferences that is well distributed across different parts of the plot. To better differentiate the consequences of each preference model, we intentionally chose a large number of points in the gray area of Figure 8, where the regret and partial return preference models would disagree (i.e., each giving a different segment a preference probability greater than 0.5).

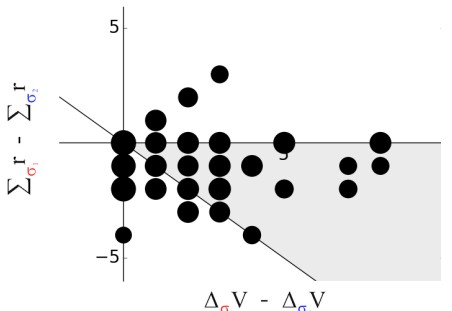

Figure 12: Coordinates from which segment pairs were sampled from during the first stage of data collection. The $x$-axis is state value differences between the two segments and the $y$-axis is partial return differences between the two segments. The areas of the circles are proportional to the number of samples at that point, and the proportionality is consistent across this plot and the 3 subplots of Figure 13.

We now describe our segment-pair sampling process more specifically. We first we constructed all unique segments of length 3 and then exhaustively paired them, resulting in nearly 30 million segment pairs. Each segment pair's partial returns, start-state values, end-state values place the segment pair on a coordinate in Figure 8, and segment pairs that are not on any of the dots in Figure 8 were discarded. For the segment pairs at each coordinate, we further divided them into 5 bins: non-terminal segments with the same start state and different end states, non-terminal segments with different start states and different end states, terminal segments with the same start state and same end state, terminal segments with a different start states and the same end state, and bin of segment pairs that fit in none of the other bins. Segment pairs in the 5th bin were discarded. From each of the 4 bins corresponding to each point in Figure 8, we randomly sampled 20 segment pairs. If the bin did not have at least 20 segment pairs, all segment pairs in the bin were "sampled". All sampled segment pairs from all bins for all points in Figure 8 made up the pool of segment pairs used with Mechanical Turk. For each subject, 50 segment pairs were randomly sampled from this pool. We gathered data until we had roughly 20 labeled segment pairs per bin. After filtering subject data, this first stage contributed 1437 segment pairs out of the 1812 pairs used in our reward learning experiments in Section 6.3 and Appendix F.3.

**Second stage** When we conducted the reward-learning evaluation in Section 6 with only the data from the first stage, $P_{\Sigma r}$ performed very poorly, always performing worse than uniformly random. This performance difference is shown in Appendix F.3. In contrast $P_{regret}$ performed well, always achieving near-optimal performance. To better assess $P_{\Sigma r}$, we investigated its results with synthetic preferences in detail and speculated that two types of additional segment pairs would aid its performance. The first of these two types include one segment that is terminal and one that is non-terminal, which we expected to help differentiate the reward for reaching terminal states from that of reaching non-terminal ones.

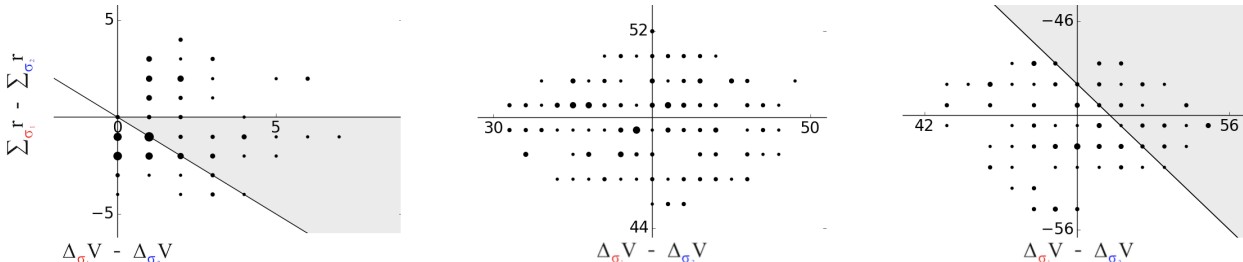

Figure 13: Coordinates from which segment pairs were sampled from during the second stage of data collection. The points are in 3 distant clusters, so they are presented in 3 separate subplots for readability. The areas of the circles are proportional to the number of samples at that point, and the proportionality is consistent across these 3 subplots and Figure 12.

The second of these two types are two segments that each terminate at different $t$ values. For example, one segment terminates on its end state, $s^\sigma_{|\sigma|}$, and another terminates after its first transition, at $s^\sigma_1$. These early-terminating segments can be viewed either as shorter segments or as segments of the same length as the other segments ($|\sigma| = 3$), where they reach absorbing state from which no future reward can be received. We speculated that this second type of segment pairs would help learn the negative reward component for each move (i.e., the gas cost). Specifically, in the first stage's data, both segments in a pair always have the

same number of non-terminal transitions, seemingly preventing preferences from providing information about whether an extra transition (from non-absorbing state) generally resulted in positive or negative reward. These segment pairs were included in all results unless otherwise stated. Note that this second type addresses the identifiability issue of the partial return model related to a constant shift in the reward function and discussed in Section 3.2.2 and Appendix F.2.2. The speculation described above was a conceptual predecessor of our understanding of this identifiability issue. In particular, any change to this gas cost—which is given at every time step—is equivalent to a constant shift in the reward function.

We now describe our segment-pair sampling process for the second stage more specifically. For the first additional type of segment pair, where one segment is terminal and one is not, we randomly pair terminal and non-terminal segments from the first-stage pool of segment pairs drawn from to present to subjects. In this pairing, each segment is only used once, and pairing stops when one of all terminal segments or all non-terminal segments have been paired. The corresponding coordinates for these pairs are shown in the two right most plots of Figure 13. For the second additional type of segment pair, we utilize all terminal segments from the pool of segment pairs shown to subjects in the first stage. For each of these terminal segments, we construct two additional segments: one that shifts the segment earlier, removing the first state and action and adds a dummy transition within absorbing state at the end, and another that shifts the segment two timesteps earlier and adds two such dummy transitions at the end. These two newly constructed segments are then each paired with the original segment, producing two new pairs for each terminal segment in the data set. The corresponding coordinates for these segment pairs are shown in the left most plot of Figure 13.

All of both types of additional segments pairs are then characterized by the coordinates shown in Figure 13. Then, as with the first stage, we randomly sampled 20 segment pairs from each coordinate to make the experimental pool for the second round of Mechanical Turk data collection. If 20 segment pairs were not available at a coordinate, we used all segment pairs for that coordinate. As in the first stage, 50 segment pairs were randomly sampled from this pool to be presented to each subject during preference elicitation. After filtering subject data, this first stage contributed 311 segment pairs out of the 1812 pairs used in our reward learning experiments in Section 6.3 and Appendix F.3.

### D.4  The study design pattern

This work follows an experimental design pattern that is often used for studying methods that take human input for evaluating the desirability of behaviors or outcomes. This pattern is illustrated for the specific case of learning reward functions from preferences in Figure 9. In this pattern, human subjects are taught to understand a specific task metric and/or are incentivized to align their desires with this metric. The human subjects then provide input to some algorithm that has no knowledge of the performance metric, and this algorithm or learned model is evaluated on how well its output performs with respect to the hidden metric. For another example, see Cui et al. (2021).

## E   Descriptive results

### E.1   Derivation of $regret_{\mathbf{d}}(\sigma_2|\tilde{r}) - regret_{\mathbf{d}}(\sigma_1|\tilde{r}) = (\Delta_{\sigma_1} V_{\tilde{r}} - \Delta_{\sigma_2} V_{\tilde{r}}) + (\Sigma_{\sigma_1}\tilde{r} - \Sigma_{\sigma_2}\tilde{r})$

The derivation below supports our assertion in the first paragraph of Section 5.1.

$regret_{\mathrm{d}}(\sigma_2|\tilde{r}) - regret_{\mathrm{d}}(\sigma_1|\tilde{r})$

$$
\begin{aligned}
&= \Big( [V_{\tilde{r}}^*(s_0^{\sigma_2}) - (\Sigma_{\sigma_2}\tilde{r} + V_{\tilde{r}}^*(s_{|\sigma_2|}^{\sigma_2}))] - [V_{\tilde{r}}^*(s_0^{\sigma_1}) - (\Sigma_{\sigma_1}\tilde{r} + V_{\tilde{r}}^*(s_{|\sigma_1|}^{\sigma_1}))] \Big) \\
&= \Big( [V_{\tilde{r}}^*(s_0^{\sigma_2}) - V_{\tilde{r}}^*(s_{|\sigma_2|}^{\sigma_2})] - [V_{\tilde{r}}^*(s_0^{\sigma_1}) - V_{\tilde{r}}^*(s_{|\sigma_1|}^{\sigma_1})] \Big) - \Big( \Sigma_{\sigma_2}\tilde{r} - \Sigma_{\sigma_1}\tilde{r} \Big) \\
&= \Big( [V_{\tilde{r}}^*(s_{|\sigma_1|}^{\sigma_1}) - V_{\tilde{r}}^*(s_0^{\sigma_1})] - [V_{\tilde{r}}^*(s_{|\sigma_2|}^{\sigma_2}) - V_{\tilde{r}}^*(s_0^{\sigma_2})] \Big) + \Big( \Sigma_{\sigma_1}\tilde{r} - \Sigma_{\sigma_2}\tilde{r} \Big) \\
&= (\Delta_{\sigma_1} V_{\tilde{r}} - \Delta_{\sigma_2} V_{\tilde{r}}) + (\Sigma_{\sigma_1}\tilde{r} - \Sigma_{\sigma_2}\tilde{r})
\end{aligned}
\tag{15}
$$

### E.2   Losses of an expanded set of preference models on the human preferences dataset

Table 3 shows an expansion of Table 1, including models introduced in Appendix B. The logistic linear preference model, $P_{log-lin}$, provides a lower bound, given that it can express either $P_{regret}$ or $P_{\Sigma r}$ and that we do not not observe any overfitting of its 3 parameters.

Including the constant probability of a uniformly random response, as in Christiano et al. (2017), also increases the expressivity of the model. The final three results in Table 3 show the best test loss achieved across different training runs that differ by initializing $logistic(c)$ of this model to

Table 3: Expanding on Table 1, mean cross-entropy test loss over 10-fold cross validation (n=1812) from predicting human preferences. Lower is better.

| Preference model | Loss (n=1,812) |
|---|---|
| $P(\cdot)=0.5$ (uninformed) | 0.693 |
| $P_{\Sigma r}$ (partial return) | 0.620 |
| $P_{regret}$ (regret) | 0.573 |
| $P_{log-lin}$ (logistic linear) | 0.548 |
| $P_{\Sigma r}$ with prob of uniform response | 0.620 |
| $P_{regret}$ with prob of uniform response | 0.573 |
| $P_{log-lin}$ with prob of uniform response | 0.548 |

be 0.01, 0.1, or 0.5. Surprisingly, no benefit is observed from including a constant probability of a uniformly random response. Because this augmentation of our models appears to have no effect on the likelihood, we do not include it in further analysis.

Across the 10 folds, the mean weights learned for $P_{log-lin}(\cdot|\tilde{r})$ were $\vec{w}=\langle-0.18,0.34,0.32\rangle$, where each weight applies respectively to segments' start state values, partial returns, and end state values. Scaled to have a maximum weight of 1 for easy comparison with $P_{\Sigma r}$ and $P_{regret_d}$, $\vec{w}_{scaled}=\langle-0.53,1.0,0.94\rangle$. First, we note that these weights are closer to those that make $P_{log-lin}=P_{regret_d}$ (i.e., $\vec{w}=\langle-1,1,1\rangle$) than to those that make $P_{log-lin}=P_{\Sigma r}$ (i.e., $\vec{w}=\langle0,1,0\rangle$). Also, the notable deviation from the weights of $P_{regret_d}$ is the weight for the start state value, which has half as much impact as the regret preference model gives it. In other words, this lower weight suggests that our subjects did tend to weigh the maximum possible expected return from each segment's start state, but they did so less than they weighed the reward accrued along each segment and the maximum expected return from each segment's end state.

## F    Results from learning reward functions

This section provides additional implementation details for Section 6, discussion of potential improvements, and additional analyses that thematically fit in Section 6.

### F.1    An algorithm to learn reward functions with $regret(\sigma|\hat{r})$

We describe below additional details of our instantiation of Algorithm 1.

**Doubling the training set by reversing preference samples**    Because the ordering of preference pairs is arbitrary—i.e., $(\sigma_1 \prec \sigma_2) \Longleftrightarrow (\sigma_2 \succ \sigma_1)$—for all preference datasets we double the amount of data by duplicating each preference sample with the opposite ordering and the reversed preference. This provides more training data and avoids learning any segment ordering effects.

**Collecting the policies from which successor feature functions are calculated**    For this article's instantiation of Algorithm 1, we collect successor feature functions by randomly sampling a large number of reward functions and then calculating the successor feature functions for their optimal polices. This procedure is more precisely described below.

1. Create a reward function by sampling with replacement each element of its weight vector, $\boldsymbol{w_{\tilde{r}}}$, from $\{-50,-10,-2,-1,0,1,5,10,50\}$.

2. For this reward function, use value iteration to approximate its maximum entropy optimal policy and that policy's successor feature function.

3. If this successor feature function policy differs from all previously calculated successor feature functions, save it and go to step 1.

4. Otherwise it is a redundant policy. If less than 300 consecutive redundant policies have been found, go to step 1.

The policy collection process above is terminated after 300 consecutive redundant policies are found. Finally, we calculate the maximum entropy optimal policy for the optimal policy for the ground-truth reward function, $r$, and *remove the successor feature function for any policy that matches the optimal policy for $r$*. In other words, we remove any policies for other reward functions that were also optimal for $r$, making the regret-based learning problem more difficult. We ensured that the ground-truth reward function was not represented to better approximate real-world reward learning applications, in which one would be unlikely to have the optimal policy for learning a successor features function. On the specific delivery task on which we gathered human preferences, the process above resulted in 70 reward functions.

**Early stopping without a validation set**    During training, the loss for the $P_{regret}$ model tended to show cyclical fluctuations, reaching low loss and then spiking. To handle this volatility, we used the $\hat{r}$ that achieved the lowest loss over all epochs of training, not the final $\hat{r}$. For $P_{\Sigma r}$ and $P_{regret}$, we found no evidence of overfitting with our linear representation of the reward function, but with a more complex representation, such early stopping likely should be based upon the loss of the model on a validation set. A better understanding of the cyclical loss fluctuations we observe during training could further improve learning with $P_{regret}$.

**Discounting during value iteration**    Despite the delivery domain being an episodic task, a low-performing policy can endlessly avoid terminal states, resulting in negative-infinity values for both its return and successor features based on the policy. To prevent such negative-infinity values, we apply a discount factor of $\gamma = 0.999$ during value iteration—which is also where successor feature functions are learned—and when assessing the mean returns of policies with respect to the ground-truth reward function, $r$. We chose this high discount factor to have negligible effect on the returns of high-performing policies (since relatively quick termination is required for high performance) while still allowing value iteration to converge within a reasonable time.

**Hyperparameters for learning**    Below we describe the other specific hyperparameters used for learning a reward function with both preference models. These hyperparameters were used across all experiments. For all models, the learning rate, softmax temperature, and number of training iterations were tuned on the noiseless synthetic preference data sets such that each model achieved an accuracy of 100% on our specific delivery task and then were tuned further on stochastic synthetic preferences on our specific delivery task.

*Reward learning with the partial return preference model*
learning rate: 2; number of training epochs: 30,000; and optimizer: Adam (with $\beta_1 = 0.9$ and $\beta_2 = 0.999$, and eps= $1e-08$).

*Reward learning with the regret preference model*
learning rate: 0.5; number of training epochs: 5,000; optimizer: Adam (with $\beta_1 = 0.9$, $\beta_2 = 0.999$, and eps=$1e-08$); and softmax temperature: 0.001.

*Logistic regression with both preference models, for the likelihood analysis in Section 5.2 and Appendix E.2*
learning rate: 0.5; number of training iterations: 3,000; optimizer: stochastic gradient descent; and evaluation: 10-fold cross validation.

**Computer specifications and software libraries used**    The computer used to run experiments shown in Figures 15,16,17,18, 19, 21, 22, and 23 had the following specification. Processor: 2x AMD EPYC 7763 (64 cores, 2.45 GHz); Memory: 284 GB.

The computer used to run all other experiments had the following specification. Processor: 1x Core™ i9-9980XE (18 cores, 3.00 GHz) & 1x WS X299 SAGE/10G | ASUS | MOBO; GPUs: 4x RTX 2080 Ti; Memory: 128 GB.

Pytorch 1.7.1 (Paszke et al., 2019) was used to implement all reward learning models, and statistical analyses were performed using Scikit-learn 0.23.2 (Pedregosa et al., 2011).

### F.1.1    For Algorithm 1, choosing an input set of policies for learning successor feature functions

A set of policies is input to Algorithm 1 and used to create successor feature functions, which are in turn used for generalized policy improvement (GPI) to efficiently estimate optimal value functions for $\hat{r}$ during learning. Which policies to insert is an important open question for successor-feature-based methods in general, but our

intuition is that the performance of GPI under successor-feature-based methods is improved with a greater diversity of successor feature functions (via a diverse set of policies) and by having some policies that perform decently (but not necessarily perfectly) on the reward functions for which $V^*$ and $Q^*$ outputs values are being estimated via GPI.

Recalling that an input policy can come from policy improvement with an arbitrary reward function, we offer the following ideas for how to source an input set of policies.

- Choose reward function parameters according to some random distribution (as we do).

- Create a set of reward functions that differ in a structured way, such as each reward function being a point in a grid formed in parameter space.

- Learn policies from a separate demonstration dataset, using an imitation learning algorithm.

- Bootstrap from $\hat{r}$. Specifically, during learning augment the current set of successor feature functions ($\Psi_Q^{\pi_{SF}}$ and $\Psi_V^{\pi_{SF}}$) by learning one new successor feature function via policy improvement on the current $\hat{r}$; then continue reward learning with the augmented set of successor feature functions and repeat this process as desired.

The input set of policies could come from multiple different sources, including the ideas above.

### F.1.2   Instantiating Algorithm 1 for reward functions that may be non-linear

Algorithm 1 operates under the assumption that the reward function can be represented as a linear combination of reward features. These reward features are obtained through a reward-features function $\phi$, which is given as input to the algorithm. Here we address situations when the linearity assumption does not hold.

If the state and action space are discrete, one could linearly model all possible (deterministic) reward functions by creating a feature for each (s,a,s') that is 1 for (s,a,s') and 0 otherwise. A downside of this approach is that the learned reward function cannot benefit from generalization, which has two negative consequences. First, in complex tasks, generalization would typically have decreased the training set size required to learn a reward function $\hat{r}$ with optimal policies that perform well on the ground-truth reward function, $r$. Second, the reward function will not generalize to different tasks that share the same *symbolic* reward function, such as always having the same reward from interacting with a particular object type.

If the reward features are unknown or the reward is known to be non-linear, another method is to create a reward features function that permits a linear *approximation* of the reward function. Several methods to derive some or all of these reward features appear promising:

- Reward features can be learned by minimizing several auxiliary losses in a self-supervised fashion, as by Brown et al. (2020). After optimizing for these various objectives using a single neural network, the activations of the penultimate layer of this network can be used as reward features. Such auxiliary tasks may include minimizing the mean squared error of the reconstruction loss for the current state from a lower-dimensional embedding and the original state, predicting how much time has passed between states by minimizing the mean squared error loss (i.e., learning a temporal difference model), predicting the action taken between two states by minimizing the cross entropy loss (i.e., learning an inverse dynamics model), predicting the next state given the current state and action by minimizing the mean squared error loss(i.e., learning a forward dynamics model), and predicting which of two segments is preferred given a provided ranking by minimizing the t-rex loss.

- Reward features could also be learned by first learning a reward function represented as a neural network *using a partial return preference model*, and then using the activations of the penultimate layer of this neural network to provide reward features.

### F.2 Results from synthetic preferences

#### F.2.1 Learning reward functions from 100 randomly generated MDPs

Here we describe how each MDP in the set of 100 MDPs discussed in section 6.2 was generated. We also extend the analysis to illustrate how often each preference model performs better than uniformly random and give further details on our statistical tests.

**Design choices** The 100 MDPs are all instances of the delivery domain, but they have different reward functions. The height for each MDP is sampled from the set {5,6,10}, and the width is sampled from {3,6,10,15}. The proportion of cells that are terminal failure states is sampled from the set {0,0.1,0.3}. There is always exactly one terminal success state. The proportion of "mildly bad" cells were selected from the set {0,0.1,0.5,0.8}, and the proportion of "mildly good" cells were selected from {0,0.1,0.2}. Mildly good cells and mildly bad cells respectively correspond to cells with coins and roadblocks in our specific delivery task, but the semantic meaning of coins and roadblocks is irrelevant here. Each sampled proportion is translated to a number of cells (rounding down to an integer when needed) and then cells are randomly chosen to fill the grid with each of the above types of states until the proportions are satisfied.

Then, the ground-truth reward component for each of the above cell types were sampled from the following sets:

- Terminal failure states: {0,1,5,10,50}

- Terminal success states: {−5,−10,−50}

- Mildly bad cells: {−2,−5,−10}

Mildly good cells always have a reward component of 1, and the component for white road surface cells is always -1. There are no cells with a higher road surface penalty (analogous to the bricks in the delivery domain).

**Better than random performance** Figure 14 complements the results in Figure 10, showing the percentage of MDPs in which each preference model outperforms a policy that chooses actions according to a uniformly random probability distribution. We can see that at this performance threshold, lower than that in Figure 10, the regret preference model outperforms the partial return preference model in most conditions. Even when their performance in this plot—based on outperforming uniformly random actions—is nearly identical, Figure 10 shows that the regret preference model achieves near optimal performance at a higher proportion.

**Details for statistical tests** We performed a Wilcoxon paired signed-rank test on the normalized average returns achieved by each model over the set of 100 randomly generated MDPs. All normalized average returns below −1 were replaced with −1, so that all such returns were in the range [−1,1]. This clipping was done because any normalized average return below 0 is worse than uniformly random, so the difference between a normalized return of −1 and −1000 is relatively unimportant compared to the difference between 1 and 0. Results are shown in Table 4.

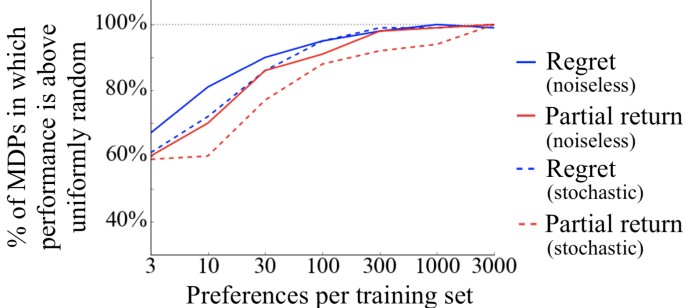

Figure 14: Comparison of performance over 100 randomly generated deterministic MDPs, showing the percentage of MDPs in which each model performed better than an agent taking actions by a uniformly random policy. This plot complements Figure 10, which shows the percentage of MDPs in which the models perform near-optimally.

Additionally, we investigate whether $P_{regret}$ and $P_{\Sigma r}$ learn near-optimal policies on the same MDPs within this set of 100 randomly generated MDPs. Results for this analysis are shown below.

Table 4: Results of the Wilcoxon paired signed-rank test on normalized average returns for each preference model.

| Preference generator type | $|D_\succ|=3$ | $|D_\succ|=10$ | $|D_\succ|=30$ | $|D_\succ|=100$ | $|D_\succ|=300$ | $|D_\succ|=1000$ | $|D_\succ|=3000$ |
|---|---|---|---|---|---|---|---|
| Noiseless ($P_{regret}$ vs. $P_{\Sigma r}$) | w=1003, p=0.115 | w=917, p=0.007 | w=739, p=0.012 | w=487, p=0.007 | w=284, p<0.001 | w=301, p=0.002 | w=289, p=0.001 |
| Stochastic ($P_{regret}$ vs. $P_{\Sigma r}$) | w=979, p=0.541 | w=1189.5, p=0.018 | w=891, p=0.027 | w=710, p=0.018 | w=285, p<0.001 | w=460, p=0.002 | w=199, p<0.001 |

Table 5: A table showing the count of the number of MDPs where both, either, or neither of the models achieved near optimal performance.

| Model(s) | $|D_\succ|=3$ | $|D_\succ|=10$ | $|D_\succ|=30$ | $|D_\succ|=100$ | $|D_\succ|=300$ | $|D_\succ|=1000$ | $|D_\succ|=3000$ |
|---|---|---|---|---|---|---|---|
| Both models | 31 | 40 | 66 | 72 | 83 | 87 | 88 |
| Only $P_{regret}$ | 20 | 26 | 17 | 18 | 14 | 8 | 8 |
| Only $P_{\Sigma r}$ | 10 | 12 | 7 | 8 | 3 | 3 | 3 |
| Neither | 39 | 22 | 10 | 2 | 0 | 2 | 1 |

### F.2.2 The effect of including transitions from absorbing state

In Section 3.2.2 we describe one context in which partial return is not identifiable because reward functions that differ by a constant can have different sets of optimal policies yet will have identical preference probabilities according to the partial return preference model (via Eqs. 2 and 9). This lack of identifiability arises specifically in tasks that have the characteristic of having at least one state from which trajectories of *different* lengths are possible. Tasks that terminate upon completing a goal or reaching a failure state typically have this characteristic. Below we describe an imperfect method to remove this lack of identifiability. Then we show that the partial return preference model does much worse in our main experiments with synthetic preferences and human preferences when this method is not applied.

**An imperfect method to prevent this source of unidentifiability for partial return** Put simply, the approach to prevent all constant shifts of reward from resulting in the same preference probabilities is to force $\hat{r}(s,a,s')=0$ in at least one tuple of state, action, and next state, $(s,a,s')$, and include in the training dataset one or more segments with that tuple.

Technically, this solution addresses the source of this identifiability issue, that any constant shift in the output of the reward function will not change the likelihood of a preferences dataset (but can change the set of optimal policies). With this solution, a constant shift cannot be applied to all outputs, since at least one reward output is hard-coded to 0. And applying a constant shift to all other outputs would change the likelihood of the infinite, exhaustive preferences dataset that is assumed in identifiability theory.

More intuitively, setting one $\hat{r}(s,a,s')=0$ for one $(s,a,s')$ tuple anchors the reward function. To explain, reward function inference effectively learns an ordering over $(s,a,s')$ tuples. Let us arbitrarily assume this ordering is in ascending order of preference. Then setting the

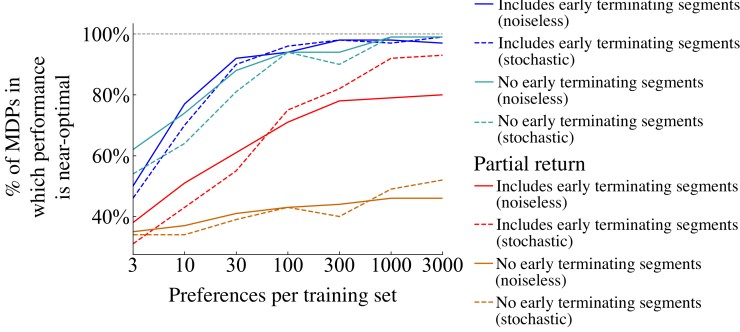

Figure 15: Performance comparison over 100 randomly generated deterministic MDPs. The results in this plot expand upon the experiments in Figure 10, adding results for datasets that do not have any segments that terminate early.

reward for one such tuple to 0 forces all $(s,a,s')$ tuples that are later in the ordering to have positive reward and all $(s,a,s')$ tuples that are earlier in the ordering to have negative reward. $(s,a,s')$ tuples that are equal in the ordering are assigned a reward of 0.

However, assuming that reward is 0 in any state has at least two undesirable properties. First, it requires adding human task knowledge that is beyond what the preferences dataset contains, technically changing the learning problem we are solving. Second, while it resolves some ambiguity regarding what reward function has the highest likelihood, this resolution may not actually align with the ground-truth reward function. In an attempt to reduce the impact of these undesirable properties, we only set the reward for *absorbing* state to 0. Absorbing state is the theoretical state that an agent enters upon terminating the task. All actions from the absorbing state merely transition to the absorbing state.

In practice, to include segments with transitions from absorbing state, we add *early terminating* segments, meaning that termination in an $n$-length segment occurs before the final transition.

The method above is not a perfect fix, however. Assuming that transitions from absorbing state have 0 reward also consequently assumes that humans consider time after termination to have 0 return. We are uncertain that humans will always provide preferences that are aligned with this assumption. For instance, if termination frees the agent to pursue other tasks with positive reward, then we might be mistaken to assume that humans are giving preferences as if there is only 0 reward after termination.

As mentioned in Section 3.2.2, past authors have acknowledged this issue with learning reward functions under the partial return preference model, apparently unaware of this solution of including transitions from absorbing states. Instead, they have used normalization (Christiano et al., 2017; Ouyang et al., 2022), tanh activations (Lee et al., 2021a), and L2 regularization (Hejna & Sadigh, 2023) that resolve their algorithms' insensitivity to constant shifts in reward. However, these papers do not discuss what assumptions regarding alignment are implicitly made by these ambiguity-resolving choices. Curiously, artificially forcing episodic tasks to be fixed horizon is common (e.g., as done by Christiano et al. (2017, p. 14) and Gleave et al. (2022)) but has not, to our knowledge, been justified as a way to address this identifiability issue. Our solution above involving early termination could be cast as a specific form of forcing a fixed horizon (infinite in this case). In one recent analysis of reward identifiability, Skalse et al. (2022) make assumptions that are identical to our solution above but do not discuss their assumptions' relationship to the partial return preference model otherwise being insensitive to constant shifts in reward when the lengths of segments in pairs are the same. We advise researchers, when forcing a fixed horizon via other methods, to explain what bias they are introducing regarding the set of optimal policies, either for generating synthetic preferences or during reward function inference.

**Empirical results** Figure 15 expands upon the synthetic-preferences results of Figure 10 in Section 6.2, adding results for datasets that lack segments that terminate early. For each combination of preference model and whether early terminating segments are included (i.e., for each different color in Figure 15), we used a learning rate chosen from testing on a set of 30 MDPs. Specifically, we chose the learning rate from the set {0.005,0.05,0.5,1,2} that had the highest mean percentage of near-optimal performance on these 30 MDPs across training sets of 30, 300, and 3000 preferences. These 30 MDPs were taken from the 100 randomly generated MDPs used in Section 6.2. So, for testing with the chosen learning rates, we replaced them with 30 new MDPs generated by the same sampling procedure, determining the 100 MDPs used for these results. Also, a different random seed is used here than in Section 6.2. In these results shown in Figure 15, performance by the partial return model is worse without early terminating segments, whereas the regret model does not appear to be sensitive to their inclusion.

The effect of removing such early-terminating segments from the *human* preferences dataset is tested in Appendix F.3.4. There we similar results: only performance with the partial return model is harmed by removing segment pairs with early-terminating segments and the decrease in performance is severe.

### F.2.3 Varying segment length

Here we consider empirically the effect of different fixed lengths of the segments in the preference data set. All other experiments in this article assume the segment length $|\sigma|=3$, and this analysis is a limited investigation into whether changing $|\sigma|$ affects results. In general, we suspect that increasing segment length has a trade off:

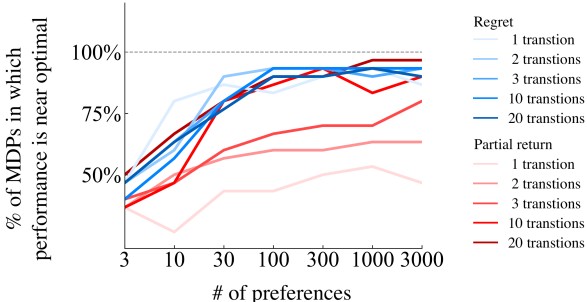

Figure 16: Using noiselessly generated preferences, comparison of performance over 100 randomly generated deterministic MDPs, showing the percentage of MDPs in which each model performed near-optimally after learning from preference datasets of different segment lengths.

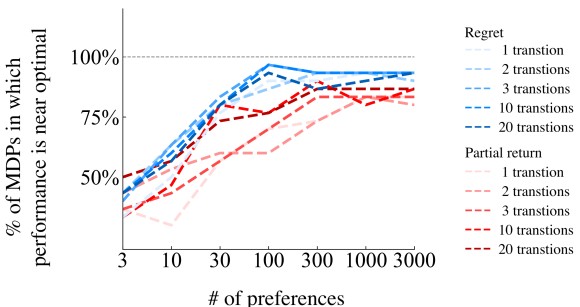

Figure 17: Using stochastically generated preferences, comparison of performance over 100 randomly generated deterministic MDPs, showing the percentage of MDPs in which each model performed near-optimally after learning from preference datasets of different segment lengths.

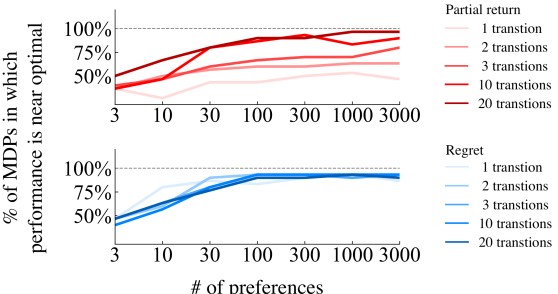

Figure 18: The results from Figure 16 (noiselessly generated preferences) divided by preference model, allowing further perspective.

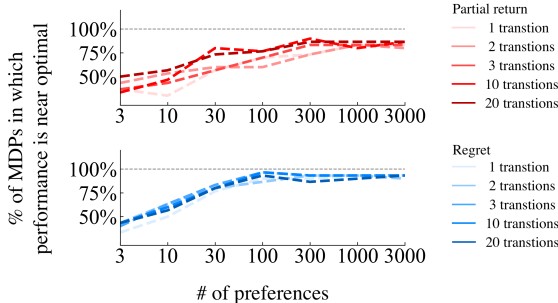

Figure 19: The results from Figure 17 (stochastically generated preferences) divided by preference model, allowing further perspective. Segment lengths 4 and 6 are added here.

it makes credit assignment within the segment more difficult yet also gives preference information about more unique and complex combinations of events, which could reduce the probability of the preference providing new information that is not already contained in the remainder of the training dataset. Put more simply, preferences over larger segments appear to give information that is harder to use but covers more transitions.

To conduct this analysis, the following process is followed for each of 30 MDPs that were sampled as described in Appendix F.2.1. For each $n \in \{1,2,3,4,6,10,20\}$, we synthetically create preference datasets with segment length $|\sigma| = n$. Each segment was generated by choosing a non-terminal start state and $n$ actions, all uniformly randomly. As in Appendix F.2.1, each preference model acts as a preference generator to label these segment pairs, resulting in datasets that differ only in their labels, and then each preference model is used for reward learning on the same dataset it labeled.

We observe the following:

- With noiselessly generated preferences (Figure 16), the performance with each preference model is similar for segments with 20 transitions, though it is sometimes slightly better with the partial return preference model. At a segment length of 10, performance with the regret preference model is better or similar, depending on the number of preferences. For all other segment lengths, performance with the regret preference model is better.

- With stochastically generated preferences (Figure 17), the performance with each preference model is generally better with the regret preference model, although sometimes the partial return preference model has marginally better performance. Additionally, the variance of performances with the partial return preference model is higher.

Additionally, we conduct Wilcoxon paired signed-rank tests for a positive effect of segment length on performance. Four tests are conducted, one per combination of preference model and whether preferences are generated noiselessly or stochastically. The steps to conduct this test are below. For each combination of preference model and each preference dataset of size $|D_\succ| \in \{3,10,30,100,300,1000,3000\}$, we calculate Kendall's $\tau$ correlation measures between the following two orderings:

- segment lengths in ascending order, (1,2,3,10,20), and

- segment lengths in the ascending order of the corresponding percentage of MDPs in which near-optimal performance was achieved (e.g., if performance is near-optimal in 90% of MDPs for $|\sigma| = 20$ and is near-optimal in 60% of MDPs for $|\sigma| = 10$, then $|\sigma| = 20$ would be later in the ordering than $|\sigma| = 10$).

For each of the resultant 7 $\tau$ values—one for each $|D_\succ|$—we create a pair for a Wilcoxon paired signed-ranked test: $(\tau, 0)$. The 0 in the pair is the expected Kendall's $\tau$ value for uniformly random orderings of segment lengths. Therefore, the pair represents a comparison between the correlation between a training dataset's segment length and its performance and no correlation. Each Wilcoxon paired signed-rank test is conducted on these 7 pairs.

- With noiselessly generated preferences and the partial return preference model (Figure 18), $p = 0.016$ and the mean $\tau$ across the 7 training dataset sizes is 0.80, indicating a significant and large correlation between segment length and ordering by performance. Our visual inspection of Figure 18 shows that the effect size is large, so we conclude that in this experiment increasing segment length meaningfully improves performance.

- With stochastically generated preferences and the partial return preference model and (Figure 19), $p = 0.016$ and the mean $\tau$ across the 7 training dataset sizes is 0.51, indicating a significant and moderate correlation between segment length and ordering by performance. Our visual inspection of Figure 19 shows that the effect size is smaller but still observable, so we conclude that in this experiment increasing segment length somewhat improves performance.

- With noiselessly generated preferences and the regret preference model (Figure 18), $p = 0.219$ and the mean $\tau$ across the 7 training dataset sizes is 0.25, which is not a significant between segment length and ordering by performance. Likewise, our visual inspection of Figure 19 does not reveal any effect, so we conclude that in this experiment increasing segment length does not improve performance.

- With stochastically generated preferences and the regret preference model (Figure 19), $p = 0.016$ and the mean $\tau$ across the 7 training dataset sizes is 0.47, indicating a significant and moderate correlation between segment length and ordering by performance. Our visual inspection of Figure 19 does not reveal this effect, indicating that the effect size is small, so we conclude that in this experiment increasing segment length improves performance with only minor effect.

### F.2.4  Reward learning in stochastic MDPs

Although we theoretically consider MDPs with stochastic transitions in Section 3, we have not yet *empirically* compared $P_{\Sigma r}$ and $P_{regret}$ in tasks with stochastic transitions, which we do below.

We randomly generated 20 MDPs, each with a $5 \times 5$ grid. Instead of terminal cells that are associated with success or failure, these MDPs have terminal cells that are either risky or safe. A single terminal *safe* cell was randomly placed, and the number of terminal *risk* cells was sampled from the set $\{1,2,7\}$ and then these terminal risk cells were likewise randomly placed. No other special cells were used in this set of MDPs. To add stochastic transitions, the delivery domain was modified such that when an agent moves into a terminal risk cell

Table 6: Stochastic MDPs: Proportion of 10 MDPs in which performance was near optimal, with varied reward functions. Entering a terminal risk cell results in $r_{win}$ and $r_{lose}$, each with 50% probability.

| Preference generator | $r_{win} = 1$ $r_{lose} = -50$ | $r_{win} = 1000$ $r_{lose} = -50$ | $r_{win} = 100$ $r_{lose} = -1$ | $r_{win} = 100$ $r_{lose} = -1000$ |
|---|---|---|---|---|
| Noiseless $P_{regret}$ | 1.0 | 1.0 | 1.0 | 1.0 |
| Stochastic $P_{regret}$ | 1.0 | 1.0 | 1.0 | 1.0 |
| Noiseless $P_{\Sigma r}$ | 1.0 | 0.0 | 1.0 | 0.0 |
| Stochastic $P_{\Sigma r}$ | 1.0 | 0.0 | 1.0 | 1.0 |

there is a 50% chance of receiving a lower reward, $r_{lose}$, and a 50% chance of receiving a higher reward, $r_{win}$. All other transitions are deterministic. As in the unmodified delivery domain, moving to any non-terminal state results in a reward of -1. Moving to the terminal safe state yields a reward of +50, like the terminal success state of the unmodified delivery domain. Therefore, depending on the values of $r_{win}$ and $r_{lose}$, it may be better to move into a terminal risk state than to avoid it. All segments were generated by choosing a start state and three actions, all uniformly randomly. For each MDP, the preference dataset $D_\succ$ contains 3000 segment pairs.

The 10 MDPs of each condition differed from those of the other conditions by their ground-truth reward function $r$, with different $r_{win}$ and $r_{lose}$ values. As in Section 6.2, regardless of whether the stochastic or noiseless version of preference model generates the preference labels, the stochastic version of the same preference model is used for learning the reward function.

The results are shown below in Table 6, indicating that for both noiseless and stochastic preference datasets, $P_{regret}$ is always able to achieve near-optimal performance, whereas $P_{\Sigma r}$ is not. These results expand upon and support the first proof of Theorem 3.2 in Section 3.

### F.2.5 Generating preferences and learning reward functions with different preference models

Using synthetically generated preferences, here we investigate the effects of choosing the incorrect model. Specifically, either $P_{regret}$ or $P_{\Sigma r}$ generates preference labels, and then the *other* preference model is used to learn a reward function from these preference labels. Through this mixing of preference models, we add two new conditions to the analysis in the delivery domain in Section 6.2. The results are shown in Figure 20. We observe that, as expected, each preference model performs best when learning on preference labels it generated. Of all four combinations of preference models during generation and reward function inference, the best performing combination is doing reward inference with $P_{regret}$ on preferences generated by $P_{regret}$.

Between the two mixed-model conditions, we observe that learning with the partial return preference model on preferences created via regret outperforms the reverse. These results suggest that learning reward functions with the regret preference model may be more sensitive to how much the preference generator deviates from it. However, we note that humans *do not* appear to be giving preferences according to partial return, so these results—though informative—are not worrisome.

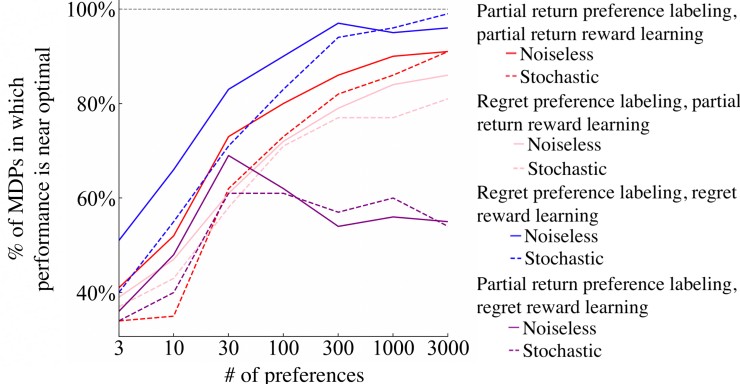

Figure 20: Performance comparison over 100 randomly generated deterministic MDPs, with synthetically generated preferences. This plot expands upon Figure 10 by including mismatches between the preference model generating preference data and the preference model used during learning of the reward function.

### F.3 Results from human preferences

In this section we expand upon the results described in Section 6.3, which involve learning reward functions from the dataset of human preferences we collected.

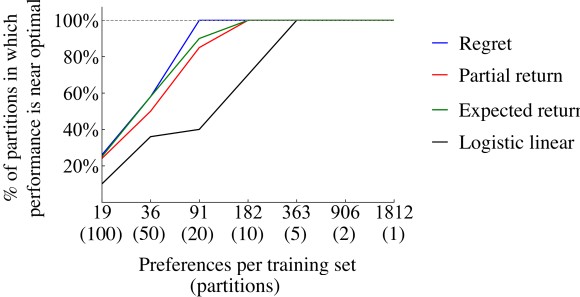

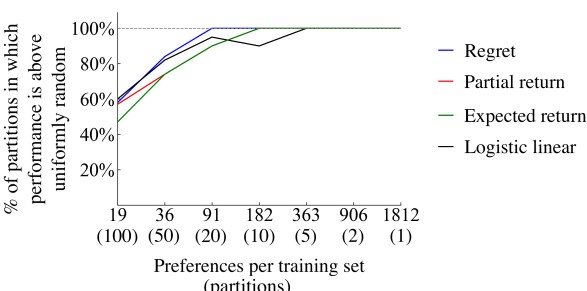

Figure 21: Performance comparison over various amounts of human preferences. Each partition has the number of preferences shown or one less. This plot is identical to Figure 11 except that results for the expected return preference model are included and a different random seed was used to partition the human preferences.

Figure 22: Performance comparison over various amounts of human preferences. Each partition has the number of preferences shown or one less. This plot tests the same learned reward functions as in Figure 21, but it thresholds on outperforming a uniformly random policy rather than on near-optimal performance.

### F.3.1 Wilcoxon paired signed-rank test

For the Wilcoxon paired signed-rank test described in Section 6.3, normalized mean returns were clipped to $[-1,1]$ as in Appendix F.2.1. The result from each test is shown in Table 7. These results are surprisingly significant, given that both models reach 100% near-optimal performance for 5 and 10 partitions in Figure 11. However, in this setting, learning a reward function with the regret preference model tends to result in a higher mean score than learning one with the partial return preference model, even when both are above the threshold for near optimal performance.

Table 7: Results from Wilcoxon paired signed-rank tests.

| | 5 partitions | 10 partitions | 20 partitions | 50 partitions | 100 partitions |
|---|---|---|---|---|---|
| $P_{regret}$ vs. $P_{\Sigma r}$ preference models | w=0 p=0.043 | w=6 p=0.028 | w=24 p=0.007 | w=216 p=0.003 | w=939 p=0.076 |

### F.3.2 Expanded plots, with the expected return and logistic linear preference models included

Figure 21 show the same results as Figure 11, but with additional results from the expected return preference model introduced in Appendix B.5 and the logistic linear preference model ($P_{log-lin}$) introduced in Appendix B.3. Figure 22 shows the same results with a different performance threshold, that of performing better than uniformly random action selection, which receives a 0 on our normalized mean return metric. The regret preference model matches or outperforms *both* other preference models in all partitionings of the human data, at both thresholds (near optimal and better than random).

### F.3.3 Performance on only human preferences from the first stage of data collection

As previously mentioned in Section D and Appendix D, when learning reward functions only from the data from the first stage of human data collection, the partial return model does worse. This first stage of data contains 1437 preferences, which is 79% of the full dataset. The specific performance of the partial return preference model on the full set of first-stage data (i.e., 1 partition) is a normalized mean return of $-12.7$, worse than a uniformly random policy. The regret preference model achieves 0.999, close to optimal performance of 1.0.

### F.3.4 Performance without early-terminating segments

Segment pairs with early-terminating segments are one type of segment pairs (of two types) that are in the second stage of data collection but not in this first stage. The identifiability issue of the partial return model related to a constant shift in the reward function—discussed in Section 3.2.2 and Appendix F.2.2—provides

sufficient justification for the low performance in Appendix F.3.3 observed without inclusion of early-terminating segments.

To more directly test the effect of early-terminating segments, we repeat the analysis in Appendix F.3.3 while only removing the segment pairs from the second stage that have early-terminating segments. We get the same normalized mean return, $-12.7$ for the partial return preference model and $0.999$ for the regret preference model. Therefore, removal of the early-terminating segments is sufficient to cause the partial return preference model to perform poorly.

### F.3.5 Generalization of learned reward functions to other MDPs with the same ground-truth reward function

We also test how well these learned reward functions generalize to other instantiations of the domain. To do so, we keep the same hidden ground-truth reward function but otherwise randomly sample MDP configurations identically as done for the analysis of Sec. 6.2, which is detailed in Appendix F.2.1. Procedurally, this analysis starts identically as that in Section 6.3, learning a reward function from randomly sampled partitions of the human preference data. Each learned reward function is then tested within the 100 MDPs. For example, for two partitions of the data, each algorithm learns two reward functions and each such $\hat{r}$ is tested on 100 MDPs. To test a reward function in an MDP, the same procedure of value iteration is used as in Sections 6.2 and 6.3 to find an approximately optimal policy, the performance of which is then tested on the MDP (with the ground-truth reward function).

The results are shown in Figure 23. The two preference models perform similarly at 100, 10, 2, and 1

Figure 23: Performance comparison of learned reward functions' generalization to 100 MDPs with the same reward function, when learned from various amounts of human preferences. Each partition has the number of preferences shown or one less.

partition(s), and otherwise the regret preference model outperforms the partial return preference model. The most pronounced differences are at 20 and 5 partitions, where the partial return preference model fails to reach near-optimal performance approximately twice as often as the regret preference model.

For each training set size, we conduct the same Wilcoxon paired signed-rank test as in Section 6.3 and Appendix F.3.1, except that for each of the 100 MDPs, we calculate the normalized mean return, and the *mean* of normalized mean returns across all 100 MDPs represents a single sample for the statistical test. Across all 7 training set sizes, no statistical significance is found ($p > 0.2$). Unlike most other analyses in this paper, we cannot here conclude superior performance from assuming the regret preference model.

