# OpenReview forum: "Models of human preference for learning reward functions"
_TMLR — Accepted by TMLR_

### Review · Reviewer_4vjA · 2023-08-08

**Summary Of Contributions:**

The paper studies the problem of modeling human preferences. A new preference model is proposed based on the regret rather than partial return. The author shows that the partial-return based model is unidentifiable, while the regret based on the negative advantage function value of the segment is identifiable.

**Audience:**

Yes

**Claims And Evidence:**

Yes

**Requested Changes:**

Please clarify all my comments above.

**Strengths And Weaknesses:**

Strength: The paper makes a first step towards understanding how dense regret helps the better align the policy with human preferences in RL. The proposed method associates the regret of each segment with human preferences, which utilizes the negated sum of an optimal policy’s advantage of each transition in the segment.

Weakeness:

- 1. My biggest concern is about how the regret can be approximated in practice. For RL problems, approximately knowing the optimal Q function up to $\epsilon$ error is equivalent to finding an $\epsilon$-optimal policy, assuming that maximizing over $Q^\star$ can be directly solved. Thus, it is unclear whether we can really utilize the regret notion for RLHF in practice.

- 2. Regarding the identifiability issue, I understand that the regret over segments provide more signal than partial return, making it identifiable under weak assumptions. However, as the authors mentioned, traditional approach focuses on maximum likelihood -- meaning that the preference model follows a certain statistical model (or precisely, the Bradley-Luce-Terry model for pairwise comparisons). If such model is well-specified and with enough samples, the MLE shall be identifiable. How shall we compare this with the regret based scenario in terms of identifiability?

- 3. I don't get the point of Section 3.2.3. If $|\sigma|=1$, meaning that it is a contextual bandit problem. Then it is expected that the discount factor does not play a role. Why would we say that this is an unidentifiability issue?

- 4. The algorithm focuses mostly on the linear reward case. Can the authors comment on how this can be generalized to neural networks?

---

> ### Author Response · Authors · 2023-11-30
> **Response to reviewer 4vjA**
>
> We thank the reviewer for their time and have addressed their questions and comments in separate OpenReview comments.

---

> ### Author Response · Authors · 2023-11-30
> **Topic: If humans do perfectly follow the Boltzmann distribution, how would identifiability be affected for each model?**
>
> In this case, both preference models are identifiable. This strong assumption of humans following the Boltzmann distribution has not been justified by human studies and we strongly caution against relying on it. **We refer the reviewer to App. C, where we discuss this topic in detail.**

---

> ### Author Response · Authors · 2023-11-30
> **Topic: I'm confused about Sec 3.2.3, in which |σ|=1 implies a bandit problem.**
>
> To clarify, |σ|=1 *does not imply a bandit task*. Any task, bandit or not, can produce a preference dataset with segments of length 1. We did look carefully for potential sources of confusion and rewrote Sec. 3.2.3 accordingly, which you can read in the uploaded revision.

---

> ### Author Response · Authors · 2023-11-30
> **Topic: The algorithm assumes a linear reward function. How could the regret preference model be used to learn reward functions that are represented non-linearly, such as with a neural network?**
>
> We also find such an extension of interest and refer the **reviewer to App. F.1.2 for a detailed discussion of the topic**. We particularly note that Brown et al. have addressed this issue in a similar context—using successor features to learn a reward function—by using the activations of the penultimate layer of a pre-trained deep neural network as the linear components, with which they apply a similar approach using successor features. Future research may find even better solutions.
>
> *Daniel Brown, Russell Coleman, Ravi Srinivasan, and Scott Niekum. Safe Imitation Learning via Fast Bayesian Reward Inference from Preferences. In International Conference on Machine Learning (ICML), pp. 1165–1177. PMLR, 2020.*

---

### Review · Reviewer_G5ch · 2023-09-28

**Summary Of Contributions:**

The paper studies how much existing ways of learning from human preferences based on reward [models] can recover optimal behavior, and under what assumptions. The idea proposed by the authors is that humans are more likely to rank (sub)trajectories based on regret than rewards cumulated in that segment, which is what current reward modelling paradigms assume.

In the theoretical part, the authors show that regret-based models of human preferences allow to recover optimal behaviors within standard workflows (preferences -> value function -> policy) but in some non-trivial cases models involving rewards cumulated on the subtrajectory cannot. The authors also propose an algorithm based on successor features to learn the optimal value function for practical cases where it is not available.

In the experiments, the authors study two claims: 1) whether humans do seem to follow more a regret-based model than a cumulated rewards-basesd model, 2) whether learning from human preferences works better with a regret-based model, and 3) how much we lose if the preferences actually follow a cumulative reward model compared to a regret-bsaed model because of non-identifiability. The experiments overall support the claims of the authors, with a little caveat for the last experiment where eventually "partial return" catches up with "regret"

**Audience:**

Yes

**Claims And Evidence:**

Yes

**Requested Changes:**

nothing requested per se but I would encourage the authors to address my questions on the presentation

**Strengths And Weaknesses:**

Strengths:
I really like the paper. As far as I know the way the authors address the problem of learning from preferences hasn't been addressed either in the "preference learning" literature nor in the "RL" literature (and still not in the most recent "RLHF"/LLM literature).

I think the authors make an interesting point between rewards cumulated over small trajectories vs regret (vs rewards comulated over entire trajectories). The claim coes in two parts (that's what humans do + that's the model we should use when learning from human rewards), which are validated independently.

The theoretical part is nice. I agree with the authors that in the current format, keeping the proofs in the body of the paper (instead of putting them to Appendix, which tends to be more standard nowadays) is useful as the actual insights are in there.

The algorithm is a nice-to-have more than a game changer, but somehow makes the paper feel more complete

The experiments are carried out thoroughly. The various steps are well-detailed and my feeling is that they support the claims sufficiently well for the sake of this paper.

Weaknesses:
To me the main weakness is that the paper may miss its audience. Considering the huge interests and investments in RLHF for LLMs and foundation models in general, it seems to me that the value of this paper would lie in how the insights translate for these particular applications. I think that the paper, as insightful as it can be, doesn't really address the specific LLM setting in enough depth to be directly applicable.

On the other hand, I appreciate that the application of regret vs patial return idea to RLHF in LLMs is a whole different endeavour that would likely require an entirely new paper. So I'm happy with the contribution as it stands and I think that "leaving this for future work" is reasonable.

The second main weakness from my perspective is presentation -- but this again may only be a matter of perspective. It seems to me that the presentation is directed towards a theory-inclined audience that is familiar with RL and concepts such as regret, which IMHO makes the paper hard to read for more applied researchers working on RLHF. Also, from the write-up, it is hard to see how much the results are corner cases or actual phenomena that we are likley to occur. I understand that only experience can answer the question, but I would have liked some discussion of what the authors would expect on RLHF-types of tasks, or a list of the main unknowns that should be answered in that context.

other questions/comments:
- the notion of regret (Eq 3) is based on the optimal policy but most of the value of this definition is when the trajectory doesn't end in a terminal state. At this stage of the paper I wonder what happens if we constrain the (sub)trajectories to end up in terminal states
- I am not sure to understand why identifiability (Def 3.1) should be limited to compare segments of the same length. Also, I think it should be said more clearly (now it's hidden in the middle of the definition). I missed it in a first read which made me struggle at the start of section 3.2.2
- the paragraph "regret as model for human preference" right before section 3 discussed the assumptions underlying the regret model. This paragraph is definitely nice to have, but I would also have liked some discussion on the differences in assumptions between the reward and the regret model regarding human data collection.
- Figure 2: it feels strange to me to compare the two segments since they do not  have the same starting state. As a human, it is unclear which one is "better" without doing an evaluation of each segment irrespective of the other (segment one is bad because is goes from safe to unsafe apparently because of the driver, while segment two seems good because it goes from unsafe to safe), which kills the point of comparisons in the first place. As far as I can see, RLHF as used in LLMs always focus on the same starting state, so I would encourage the authors to make sure their examples also follow this case. In particular, to me it is not clear if the human assessment should assess the starting state -- in which case I prefer to start from a safe place than an unsafe one -- it seems easier to improve from there. My understanding is that the regret cancels out the effect of the starting state within a trajectory, which seems a pretty big deal in practice if we compare trajectories with different starting states.

---

> ### Author Response · Authors · 2023-11-30
> **Response to reviewer G5ch**
>
> We appreciate the reviewer's attention and recognition of this paper's novelty, its theoretical and empirical contributions, and the intuitive appeal of the regret preference model.
>
> We have taken care to consider and respond to the reviewer's questions and comments, each labeled as a "topic" in a separate comment here.

---

> ### Author Response · Authors · 2023-11-30
> **Topic: The most impactful application of RLHF right now is to fine-tune LLMs and other foundation models. This paper isn't written to appeal to the practitioners doing this fine-tuning work, who may not have a strong RL background.**
>
> We certainly understand this perspective. We share our thoughts below, which appear to have been somewhat anticipated by the reviewer already.
>
> We decided to focus on making this a foundational article rather than one applied to a specific domain—even one as world-changing as LLMs. In our response to QGea under "**The regret preference model leads to the same RLHF fine-tuning algorithm for LLMs...**", we briefly sketch one way the regret preference model can be applied to fine tuning. This reframing of the existing approach leaves unchanged the predominant algorithm for learning from segments of length 1 but points to a new algorithm for learning from longer segments. However, the regret preference model reframing of this fine-tuning approach does not involve learning a reward function, so we judged it out of scope for this paper.
>
> We envision the audience of _this_ paper to be RLHF researchers who have a strong background in RL and can build upon the foundation we have laid. We are confident that such subsequent work will leverage the regret preference model proposed here to provide guidance to the practitioners fine-tuning foundation models.
>
> While outside the scope of this paper, our ongoing research agenda includes building on this paper in the way the reviewer appears to desire.

---

> ### Author Response · Authors · 2023-11-30
> **Topic: What if we enforce the constraint on the preferences dataset that all trajectories reach terminal state?**
>
> This question is worthwhile to spend some time on. First, to review a bit: the regret preference model differs from the partial return preference model by adding 3 components: a segment's start state value, a segment's end state value, and a correction to remove the impact of transition luck.
>
> Of the 3 components, the impact of the end state value on human preferences appears most intuitive to those people with whom we have discussed this research. Also, App. E.2 indicates that the end state value does have a strong impact on human preferences. So we understand the appeal of removing this impact to reduce the differences in the preference models.
>
> Like the reviewer, we are also interested in testing with a variety of other datasets, including ones in which all segments are full trajectories. However, the collection of human data is an expensive and time consuming process—a process other RLHF papers rarely do even once (unless backed by large industrial resources)—that we unfortunately cannot repeat for this article.
>
> We additionally note that having all segments terminate means that, to have decent coverage of the state space in these segments, then the segments in certain tasks will need to be very long. Consider an autonomous driving setting, where a trajectory is a 30-minute drive. Having an annotator evaluate 2 30-minute drives to give one preference label would use an hour of annotator time to provide little information, so **having non-terminating segments appears to sometimes be the only reasonable choice**. Another situation in which having all segments terminate is impossible is a continuing task, a large category of tasks in RL in which termination does not occur at all. Similarly, some episodic tasks can last extremely long or even indefinitely if behavior is optimal, like Tetris and balancing cart pole, which also makes using only terminating segments questionable.

---

> ### Author Response · Authors · 2023-11-30
> **Topic: Instead of allowing start states within segment pairs to differ, should we only consider pairs with the same start states (which removes one difference between the regret and partial return preference models)?**
>
> **In real-world tasks, like those for robots, the number of states is often infinite and one does not have the ability to arbitrarily choose a start state. In such tasks, finding even a single pair of segments with the exact same start state may be impossible.**
>
> Like the reviewer, we do find it appealing to collect preferences over pairs of segments with the same start state _when possible_, but we chose to not enforce that constraint to make the work more general. We're aware that the cost of this decision is that we forgo the opportunity to shed light on the specific but common category of preference dataset when all pairs have the same start state.
>
> We note that **this generality paid off**: since forcing segments to have the same start state could be limiting (e.g., when environment resets take time), it's worthwhile to know whether start state values have any predictive value when they do differ. App. E.2 indicates that *the start state value does impact human preferences in our collected dataset*, though it is less strong than that of the end state value. **Having a dataset with segment pairs of differing start states permits this analysis, which suggests we _do_ want to collect preference datasets where each segment in a pair has the same start state _when we can_, to avoid the effect of imperfectly modeling the effect of the start state value.**
>
> On the broader topic of differing start states, we have added to Figure 1 a second illustrative example that focuses on segments with different start state values.

---

> ### Author Response · Authors · 2023-11-30
> **Topic: How does assuming the partial return preference model or the regret preference model affect human data collection?**
>
> The two preference models do not necessarily affect human data collection, since they merely model a hidden process that occurs in the minds of humans.
>
> Relatedly, however, we briefly note in our Conclusion section that one could develop prescriptive methods to train or otherwise influence annotators to conform to one preference model or another.

---

> ### Author Response · Authors · 2023-11-30
> **Topic: Can you more saliently state your assumption that segments need to be the same length? Why should the definition of identifiability be limited to compare segments of the same length?**
>
> Thank you for this comment, which led us to realize that this assumption is not universally necessary. **We have removed it from our definition of an identifiable preference model (Def. 3.1).** We discuss this topic in more detail below.
>
> There are two ways that segments can be constrained to be the same length. The weaker constraint is that each segment in a segment pair should have the same length, while different pairs might have different lengths. We state that we assume this weaker constraint first in Sec. 2.1, under "**Preference datasets**", although our theoretical results in the revised manuscript do not make this specific assumption, except for Sec. 3.2.2.
>
> Let us first discuss this weaker constraint. One reason to include it is that this constraint is typical in RLHF methods. A stronger reason, in our view, is that when there are large differences in length _within_ segment pairs, both the regret and partial return preference models act in ways that strongly violate our intuition for how humans would give preferences. To ease illustration, let's consider financial tasks in which reward and money have a positive linear relationship.
>
> **Example for the partial return preference model**. Consider a segment that lasts a year and gets \\$1 per day (for \\$365 total) and another segment that lasts a day and gets \\$100. From the partial return preference model's perspective, the first segment is better, even though its _rate_ of reward is 1/100th that of the second segment. It's hard to imagine humans agreeing with that preference.
>
> **Example for the regret preference model.** Assume optimal behavior gets \\$100 each day. Consider a segment that lasts a day and gets 0 and a segment that lasts a year and gets \\$99 per day. The total regret of the first segment is 100, and the total regret of the second segment is \\$1 x 365 = \\$365. So the regret preference model—which was designed intuitively to prefer smaller deviations from optimal behavior—would prefer the egregiously suboptimal first segment over the near-optimal but longer second segment.
>
> For both models, for preferences over segments with different lengths, we suspect that using something like the *average* reward or regret would provide more alignment than using the sum of reward or regret. However, we are unsure that it's important to allow such segment pairs. Thus, we stick with this standard constraint and leave for future work consideration of how preference models should be adapted to preferences over segments of different lengths.
>
> The stronger version of this constraint is that all segments in the entire preference dataset have the same length. This assumption is made obviously for the case when |σ|=1 always. It is also made for the second proof in Sec. 3.2.1, although that proof is redundant after the first proof in Sec. 3.2.1, and so the constraint doesn't affect the generality of Theorem 3.2.

---

### Review · Reviewer_QGea · 2023-11-21

**Summary Of Contributions:**

This paper studies the problem of how to model human preferences for learning reward functions, which can then be used in Reinforcement Learning from Human Feedback. The paper proposes a new model for human preferences, the Regret model, which models the probability of one sequence being preferred to another as a logistic function over the differences in the *regret* of the two sequences. They compare this model to the standard partial return model both theoretically and empirically. On the theoretical side, they show their model is identifiable (in the limit of infinite noiseness data using the model can recover the underlying reward function), wherease the partial return model is shown by counter-example to not be identifiable in 3 scenarios: noiseless preferences, variable-horizon tasks and segments of length 1. On the empirical side, the authors introduces a toy delivery driver gridworld MDP, and gather human preference data on segments in this setting. They show that the regret model better-models the human preference data, implying it is a better descriptive model. They present an algorithm to learn this a reward function under regret model in a linear setting using successor features, and show that this algorithm can learn reward functions from synthetic and human data better than the partial return model in terms of the size of the dataset.

**Audience:**

Yes

**Broader Impact Concerns:**

I don't have any concerns of the ethical implications of the work.

**Claims And Evidence:**

Yes

**Requested Changes:**

As mentioned in the Strengths and Weaknesses, none of these changes are critical for securing my recommendation, and are rather just changes that would strengthen the claims the authors could make in this work.

* If the authors performed experiments on other settings, particularly pre-exising environment from the literature, then that would enable them to make stronger claims about the empirical promise of this modelling strategy.
* Related to this, if the authors could present an algorithm for learning under the regret preference model that will work in more general settings (for example RLHF fine-tuning of language models), that would be beneficial.

**Strengths And Weaknesses:**

# Strengths

* The paper is clear and well-written, and easy to follow.
* The paper's claims are all carefully made and backed up by theoretical or empirical results in all places.
* The theoretical results are interesting and well-explained.
* The proposed regret model of human preferences is interesting for the community, and points to lots of future work in this space.

# Weaknesses

* While this does not make the paper not worthy of acceptance, the propose model of human preferences seems much more difficult to learn a reward function under than the partial return model. The proposed algorithm relies on a linear setting and successor features, neither of which are generally the case in the standard RLHF setting of fine-tuning language models.
* Similarly, in this setting (in particular single-turn dialogue or instruction following) the partial return and regret models are the same, so this contribution is unlikely to improve results there. I don't think the authors need to change the paper to address these two points, but I thought it was worth mentioning.
* The empirical results are somewhat limited to a single environment which is proposed by the authors, which makes it difficult to asertain how performant using this model and algorithm would be in other more realistic settings. I think the paper is still worthy of acceptance, but if the authors performed experiments on other settings, particularly pre-exising environment from the literature, then that would enable them to make stronger claims about the empirical promise of this modelling strategy.

---

> ### Author Response · Authors · 2023-11-30
> **Response to reviewer**
>
> We thank the reviewer for their time and are grateful for their affirmation of this paper's writing, theory, and potential to serve as the foundation for a significant body of future work.
>
> Although the weaknesses described by the reviewer were not particularly concerning to them, we would nonetheless like to thoughtfully respond in this thread to the comment not already addressed in our general response to all reviewers.

---

> ### Author Response · Authors · 2023-11-30
> **Topic: The regret preference model leads to the same RLHF fine-tuning algorithm for LLMs as does the partial return preference model. What impact then could it have on that application of RLHF?**
>
> Great insight! For multi-turn dialogue, which most highly publicized LLMs are currently trained to support, the "reward model" is learned under an assumption of the partial return preference model. This reward model is used in a bandit setting, which is the same as setting the discount factor to 0.
>
> If segment lengths are more than 1, then this approach quickly becomes nonsensical by ignoring all actions in the segment after the first. The regret preference model points to a different algorithm, where the learned "reward model" is actually an optimal advantage function that more intuitively extends the RLHF fine-tuning algorithm for segments of length 1. The goal of LLM fine-tuning then becomes to increase the likelihood of responses predicted to have higher optimal advantages.  \
>  \
> **In short, if the preference dataset contains segments of length 2 or greater—which might be desirable, e.g., to overcome current LLMs' tendencies to completely address a prompt in one response rather than to ask follow-up questions—the regret preference model does have an impact on the derived algorithm.** We consider this discussion out of scope for _this_ paper though, since this paper focuses specifically on learning reward functions and, under the interpretation of the regret preference model, RLHF fine-tuning is not learning a reward function but rather is learning an optimal advantage function.

---

### Author Response · Authors · 2023-11-30
**General response to our reviewers**

In addition to our individual responses to each reviewer, here we list our minor revisions since submission and address two topics of common concern.\
\
We have made the following **revisions**, which are marked in blue text in the uploaded revision for easy identification.

* Figure 1 now contains a second illustrative example that focuses on segments with *different start state values*. This example is intended to give readers the intuition that their preference can be better modeled by the regret preference model when the start states differ and also that comparing two segments with different start states can be straightforward. This additional example relates to our response to reviewer G5ch.
* In Sec. 3, we sharpened the definition of identifiability to not be specific to an operation that minimizes loss. In Secs. 3.1 and 3.2, we added general descriptions of how we prove identifiability or non-identifiability.
* In response to confusion expressed by reviewer 4vjA, we revised Sec. 3.2.3 (on identifiability when all segments are of length 1) to be clearer.
* We removed the constraint in Def. 3.1 that all segments have the same length, based on considerations provoked by reviewer G5ch28. It was unnecessary for most proofs. Specific related constraints are added for two of the proofs.

\
In this thread, we address two related topics that reviewer QGea and reviewer 4vjA each raised.

---

> ### Author Response · Authors · 2023-11-30
> **Topic: What is the consequence of the difficulty of learning reward functions with the regret preference model?**
>
> We agree that learning reward functions with the regret preference model is more difficult, since it requires repeatedly approximating Q* for the learned reward function as it changes during training. _We discuss this challenge in App. A.3._ We will not repeat what is there already, but we would like to add a couple of points.
>
> *First*, we submit that an ideal algorithm for learning reward functions from preferences should both (1) have strong alignment properties and (2) be effective in complex tasks. Our paper shows that usage of the partial return preference model is problematic with respect to its alignment properties, regardless of how easy it is to learn with. And learning reward functions with the regret preference model appears to have strong alignment properties but not _yet_ be effective in complex tasks. Future research will be needed to see if that can be changed (as discussed in App. A.3).
>
> *Second*, we make the point in App. A.3 that inverse reinforcement learning (IRL) has the same challenge of needing to solve an MDP in the inner loop of learning. Put differently, concerns about scaling reward learning with the regret preference model are valid, but this concern should not detract from the significance of our research. The foundational paper on Inverse Reinforcement Learning (IRL) by Ng et al. in 2000 faced similar concerns. Yet it has founded an active and influential subfield of AI that has produced many of its ~3600 citations. Building further upon this comparison, a demonstration dataset for IRL could instead be used directly for behavior cloning (BC), removing this issue of needing to solve an MDP in the inner loop. However, a policy is learned directly, without learning a reward function, which poses trade-offs. Similarly, because regret separates optimal behavior from suboptimal behavior, a policy could be learned directly with an algorithm derived from the regret preference model. An investigation of this promising direction is beyond the scope of this paper, however, which is focused on learning reward functions.
>
> *Ng, Andrew Y., and Stuart J. Russell. Algorithms for Inverse Reinforcement Learning. Proceedings of the Seventeenth International Conference on Machine Learning. 2000.*

---

> ### Author Response · Authors · 2023-11-30
> **Topic: Using only grid worlds limits how generally we can claim that learning with the regret preference model leads to superior performance. It would be helpful (but not necessary) for this paper to include an algorithm for learning from the regret preference model in complex tasks, along with experiments in such tasks.**
>
> We agree that adding more domains would improve the generality of our empirical results and that proposing and testing an extension of our algorithm on more complex domains would be a potent next step. In case it was missed, we do however want to highlight that most of our synthetic evaluations include 100 different MDPs.
>
> A primary focus of this paper is the scientific analysis of human preferences and preference models, which conflicts with scaling up to tasks commonly used for deep RL. Specifically, testing in more complex tasks would require coarser approximations of regret, which can make results harder to interpret. For example, if a regret-based algorithm fails, we wouldn't know whether regret is to blame or whether the approximation of regret is to blame. For this reason, we decided not to focus on deep RL or complex tasks, despite the _practical_ importance of such scalability.
>
> We believe that algorithms for human-created data should be tested with human-created data, only using synthetic data to better understand the algorithms. Gathering human datasets is expensive in terms of design and engineering (see https://bit.ly/humanprefs for our UI that trains subjects and elicits their preferences). Testing these models in other tasks with real human data would have required multiple costly human subjects studies. Our contribution of a released human dataset will make testing in multiple environments easier for future research efforts in this area.
>
> In future work on the application of regret preference models, we or other researchers will face the research and engineering task of scalability. Given that IRL has made tremendous progress in this direction and the cited Brown et al. paper has scaled an algorithm with similar needs to those of Alg 1, we are highly optimistic that the methods to scale can be developed and probably already exist (e.g., in Brown et al. and in the later part of our App F.1, under "Instantiating Algorithm 1 for reward functions that may be non-linear").
>
> *Daniel Brown, Russell Coleman, Ravi Srinivasan, and Scott Niekum. Safe Imitation Learning via Fast Bayesian Reward Inference from Preferences. In International Conference on Machine Learning (ICML), pp. 1165–1177. PMLR, 2020.*

---

> ### Author Response · Authors · 2023-12-04
> **One day remaining for discussion**
>
> We would like to kindly remind our reviewers that today is the last day for discussion. (We also acknowledge that much of the discussion period fell on a long holiday, delaying *our* own initial responses.)
>
> If there are any remaining questions or concerns, we would be happy to address them.

---

### Decision · Action_Editor_Eb1n · 2024-01-08

**Recommendation:** Accept as is

**Comment:**

This paper was a bit cursed, through no fault of the authors, as we struggled to find a full set of reviewers. Once we did, the reviewer consensus was clear, and I am satisfied that the authors have taken onboard all feedback in making changes to the paper during the discussion period. I encourage the authors to look once more over the feedback and incorporate any outstanding recommendations into their camera ready draft.

**Audience:**

This will be of interest to the LLM community, and more specifically to those researchers interested in novel RL-proximal methods for improving LLMs.

**Claims And Evidence:**

This paper proposes and evaluates a regret-based reward model derived from human preference data. The authors provide both discussion of the theoretical motivations and strengths, and empirical evidence of its successful application. The reviewers agreed that on these two grounds, the paper meets the high bar for acceptance at TMLR.